# Multiple Populations in Star Clusters

Antonino P. Milone [1,2,*] and Anna F. Marino [2,3]

1 Dipartimento di Fisica e Astronomia "Galileo Galilei", Universitá di Padova, Vicolo dell'Osservatorio 3, 35122 Padova, Italy
2 Istituto Nazionale di Astrofisica—Osservatorio Astronomico di Padova, Vicolo dell'Osservatorio 4, 35122 Padova, Italy; anna.marino@inaf.it
3 Istituto Nazionale di Astrofisica—Osservatorio Astrofisico di Arcetri, Largo Enrico Fermi, 5, 50125 Firenze, Italy
* Correspondence: antonino.milone@unipd.it

**Abstract:** We review the multiple population (MP) phenomenon of globular clusters (GCs): i.e., the evidence that GCs typically host groups of stars with different elemental abundances and/or distinct sequences in photometric diagrams. Most Galactic and extragalactic clusters exhibit internal variations of He, C, N, O, Na, and Al. They host two distinct stellar populations: the first population of stars, which resemble field stars with similar metallicities, and one or more second stellar populations that show the signature of high-temperature H-burning. In addition, a sub-sample of clusters hosts stellar populations with different heavy-element abundances. The MP origin remains one of the most puzzling, open issues of stellar astrophysics. We summarize the scenarios for the MP formation and depict the modern picture of GCs and their stellar populations along with the main evolutionary phases. We show that the MP behavior dramatically changes from one cluster to another and investigate their complexity to define common properties. We investigate relations with the host galaxy, the parameters of the host clusters (e.g., GC's mass, age, orbit), and stellar mass. We summarize results on spatial distribution and internal kinematics of MPs. Finally, we review the relation between MPs and the so-called second-parameter problem of the horizontal-branch morphology of GCs and summarize the main findings on the extended main sequence phenomenon in young clusters.

**Keywords:** globular clusters; open clusters; stars: population II

## 1. Introduction

Being among the oldest objects in the Milky Way, globular clusters (GCs) provide a privileged observation window into the Universe's infancy. They hold stellar fossil records which are used in Galactic archaeology to uncover the history of the Primordial Universe by observing the Local Universe (e.g., [1]).

For many years, GCs have served as laboratories for testing the predictions of stellar evolution models since their stars were considered the best example of a simple stellar population, a population of monometallic and coeval stars from which estimates of distances and ages can be obtained [2].

For a long time, astronomers believed that GC stars formed out of one single burst of star formation and used the GC simplicity to constrain the main properties of a stellar population. Realizing that these stellar systems host multiple stellar populations has unavoidably changed our perspective, making the presence of more than one stellar population in GCs one of the most astonishing discoveries in the field of stellar populations in recent years. Most Galactic GCs originated at redshifts higher than ∼3 [3], when only a small fraction (∼6%) of the Milky Way stellar mass was formed [4]. Hence, the origin of GCs and their stellar populations precedes the assembly of most of the Galaxy, thus shedding new light on the early Galaxy formation [5].

Historically, three observational facts have challenged the notion of GCs as simple stellar populations.

- **Chemical anomalies.** It has long been known that the chemical composition of stars in GCs is not homogeneous in the elements involved in hot H-burning, such as C, N, O, Na, Al, and in some cases Mg, Si, and K. The variations in these light elements set well-known chemical patterns, such as the O–Na/C–N, and the Mg–Al anticorrelations (see [6–9], for reviews). Stars with lower N and Na and higher C and O resemble Galactic-field stars with the same metallicity, while stars enhanced in N and Na and depleted in C and O are mostly found in GCs.
- **The second parameter of the horizontal branch** indicates the phenomenon discovered in the 1960s that GCs with similar metallicities exhibit different horizontal branch (HB) morphologies [10–12].
- **Multiple sequences in the color magnitude diagram.** Since the late 1990s, there has been growing evidence of split or broad main sequences (MSs), red giant branches (RGBs), and sub-giant branches (SGBs) in GCs [13–21]. A discovery that dates back to the past decade is that the CMDs of nearly all GCs are composed of multiple sequences that can be followed continuously, along with all evolutionary phases, from the bottom of the MS toward the RGB tip and also along the HB, the asymptotic giant branch AGB, and even the white dwarf cooling sequence [22–27].

Recently, some pioneering studies on M4 revealed that chemical anomalies, multiple photometric sequences on the CMD, and the distribution of stars along the HB are different sides of the same phenomenon [19,28], which we call multiple population phenomenon. In the following, we indicate stars with Galactic-field-like chemical composition as the first population (1P). The sample of stars enhanced in He, N, Na, and Al and depleted in C and O define one or more subsequent stellar populations, dubbed altogether as second populations (2Ps).

Understanding the origin of the multiple stellar population phenomenon is a challenge for stellar evolution, stellar nucleosynthesis, and the star formation mechanisms at high redshift. Some theoretical scenarios assume that the multiple stellar populations correspond to different generations of stars, with multiple star formation events taking place. The first burst of star formation would form stars out of pristine material. Later, one or more subsequent events would build stellar populations from the ashes of more massive first-population stars, which evolve faster (e.g., [29–32]).

However, some of the observational constraints, such as the fact that 2P stars represent the majority of stars in most clusters [24,33], are hardly reconcilable with a scenario of multiple bursts of star formation. To fit this scenario, we need to assume that proto-GCs were some order of magnitudes more massive than their progeny. Such massive clusters would contribute to the Milky-Way assembly and even possibly the cosmic reionization [5]. Because of this demanding requirement, it has also been proposed that multiple stellar populations in GCs form in a single burst of star formation, with a fraction of stars being polluted by the ejecta of more massive stars of the same generation [34,35].

As of now, we do not properly understand the origin of multiple stellar populations in GCs. In recent years, various excellent reviews have been dedicated to GCs and their stellar populations (e.g., [6–9,36,37]). Here, we discuss the state of the art of observed properties of multiple stellar populations in GCs by focusing on recent results, largely based on photometry. We also provide a brief discussion on the formation scenarios and their challenges.

The paper is organized as follows. We start discussing the main observational tools to identify and characterize multiple populations in GCs (Section 2). We then define the different classes of GCs in Section 3 and summarize the scenarios for the formation of multiple populations in Section 4. Section 5 describes the chemical composition of stellar populations, whereas Section 6 illustrates the main properties of multiple populations. Section 7 is dedicated to the second-parameter problem of the HB morphology, while Section 8 discusses the extended main sequence turn offs and split main sequences of

young clusters (ages smaller than $\sim 2\,$Gyr). Finally, in Section 9 we provide a short list of observations that in our opinion could help shed light on the multiple population phenomenon.

## 2. Observational Tools to Disentangle Multiple Populations in GCs

For a long time, spectroscopy has been the main technique to characterize GC stellar populations ([6,38,39], and references therein). In the past two decades, the synergy between complementary techniques, spectroscopy and photometry, had a crucial role in guiding the design of new and more efficient observations to explore the multiple stellar populations. After the pioneering study of M4, which revealed that stars with different light-element abundance define distinct sequences in the CMD [19], we started gaining knowledge on how to construct new effective tools to investigate the different stellar populations in GCs.

The new tools that have been introduced are based on multiband photometric techniques but also involve spectral synthesis and elemental abundance inferred from spectra, which provide the chemical key to reading the photometric diagrams. In this context, a major contribution for the construction of high-precision photometric diagrams, which are very efficient in identifying distinct stellar populations, has certainly been provided by the innovative methods of photometric data analysis developed by Jay Anderson and his collaborators (e.g., [40,41]). These methods, which are based on an effective point-spread function, have allowed obtaining high-precision photometry from images collected with the Hubble Space Telescope (*HST*) and ground-based facilities.

In the following items, we provide a summary of the basic photometric ingredients that are effective to maximize the separation between stellar populations with different chemical compositions. We describe the new diagrams that have been the key tools to identify and characterize multiple populations in a large sample of clusters.

- **Wideband ultraviolet photometry**. CMDs made with $U$-band photometry are powerful tools to identify multiple stellar populations along different evolutionary phases. In their aforementioned pivotal paper on the nearest GC, M4, Marino et al. [19] have demonstrated that it is possible to disentangle stellar populations with different chemical compositions by using wideband ground-based photometry. Indeed, stars with different abundances of carbon, nitrogen, oxygen, and sodium define distinct RGB sequences in the $U$ vs. $U - B$ CMD. The main reason for the RGB split is that the $U$ filter includes NH and CN molecular bands, whereas the $B$ filter encompasses CH bands. Hence, 2P stars exhibit fainter $U$ magnitudes and redder $U - B$ colors than 1P stars as a result of their enhanced nitrogen and depleted carbon abundances [19,42].

- **Narrowband photometry**. The Strömgren $c1$ index, originally designed to measure the Balmer discontinuity strength, is also an efficient tool to detect star-to-star variations in the strength of the CN molecular bands. Since the 1990s, it has been used for detecting multiple populations along with the RGB of various GCs [14,43,44]. The set of photometric indices designed by Jae-Woo Lee and collaborators is another outstanding tool to detect multiple populations along the RGB (e.g., [45–47]).

- **Wide color baseline**. Stellar models predict that MS and RGB stars with different helium contents but the same luminosity have different effective temperatures. Hence, helium-rich stars exhibit bluer colors than stars with pristine helium abundance ($Y \sim 0.25$) as a result of their hotter effective temperatures (e.g., [48,49]). The photometric signature of helium is due to the fact that helium alters the stellar structure, while the emerging flux is rather negligible [50]. Early discoveries of split MSs in GCs with large internal helium variations were based on CMDs made with the $m_{F555W} - m_{F814W}$ and $m_{F475W} - m_{F814W}$ colors [13,16–18,51]. Wider color baselines, such as $m_{F275W} - m_{F814W}$, are more sensitive to helium variations than colors made with optical filters, thus allowing for disentangling stellar populations with small helium differences of $\Delta Y = 0.01$ or less [52].

- **Near-Infrared photometry** is an efficient tool to identify multiple populations of M dwarfs. The F110W and F160W filters of the WFC3/NIR camera onboard *HST*, which

are similar to the J and H bands, are the most widely used filters. Indeed, the F160W band is heavily affected by absorption from various molecules that contain oxygen, including $H_2O$, while F110W photometry is poorly affected by the oxygen abundance. Since 2P stars are oxygen-depleted, they have brighter $m_{F160W}$ magnitudes and redder $m_{F110W} - m_{F160W}$ colors than the 1P [23,33].

- **Two-color diagrams and pseudo color-magnitude diagrams.** Two-color diagrams involving far-UV, UV, and optical filters are widely used to identify multiple populations along different evolutionary phases. The most used ones are the $m_{F275W} - m_{F336W}$ vs. $m_{F336W} - m_{F438W}$. F225W, F343N, and F410M bands often substitute for one or more traditional filters. The reason why these filters are efficient tools to identify MPs in GCs is that F275W (or F225W) and F336W (or F343N) passbands include OH and NH molecular bands, while F438W (or F410M) comprises CN and CH bands. As a consequence, 1P stars, which are O-rich and C-rich but N-poor, are relatively bright in F336W but are fainter than 2P stars in F275W and F438W [22,53].

  To investigate multiple populations along all evolutionary sequences together, Milone et al. [53] combined these colors to define the pseudo-colors $C_{F275W,F336W,F438W} = (m_{F275W} - m_{F336W}) - (m_{F336W} - m_{F438W})$.

  The $m_{F336W} - m_{F438W}$ vs. $m_{F438W} - m_{F814W}$ two-color diagram and the $C_{F336W,F438W,F814W} = (m_{F336W} - m_{F438W}) - (m_{F438W} - m_{F814W})$ pseudo-color, together with the analogous diagram made with ground-based photometry in U, B, and I bands are also widely used to investigate multiple populations in GCs [22,53,54].

- The photometric diagram dubbed a **Chromosome Map** (ChM) is a pseudo-two-color diagram that is built for MS, RGB, or AGB, separately [24,55]. The main difference with a simple two-color diagram is that the sequences of MS, RGB, or AGB stars are verticalized in both dimensions in such a way that stars of each stellar population are clustered in a small area of the ChM. The ChM is derived from colors that are sensitive to the specific composition of GC stars with the aim of maximizing the separation among the distinct populations. The traditional ChMs are built by combining the $m_{F275W} - m_{F814W}$ color, which is mostly sensitive to helium variations, with the $C_{F275W,F336W,F814W}$, which is mainly a proxy of nitrogen abundance[1].

  Other ChMs are constructed from colors of M dwarfs made with photometry in optical (e.g., $m_{F606W} - m_{F814W}$) and NIR bands (e.g., $m_{F110W} - m_{F160W}$) to disentangle stellar populations with different oxygen abundances [57]. Recently, a ChM that includes photometry in the F280N band has been introduced and is sensitive to the magnesium content of stellar populations [58].

- The **universal ChM** introduced by Marino et al. [59] allows for properly comparing the maps of different GCs. It differs from the ChM because its pseudo-color extension does not depend on cluster metallicity.

- **Integrated photometry.** The diagrams derived from integrated $C_{F275W,F336W,F438W}$ pseudo-color and the $m_{F275W} - m_{F814W}$ color are valuable tools to detect the multiple population properties from GC integrated light [60]. Work based on 56 Galactic GCs, where multiple populations are widely studied, revealed that after the dependence from metallicity is removed, the color residuals depend on the maximum internal helium variation within GCs and on the fraction of 2P stars. Hence, this tool has the potential to extend the investigation of multiple populations outside the local group [60].

Some photometric diagrams that allow disentangling multiple populations in GCs are plotted in Figure 1 for 47 Tucanae [23,61]. These include the $m_{F814W}$ vs. $C_{F275W,F343N,F438W}$ and the $I$ vs. $C_{U,B,I}$ diagrams (upper panels), the $m_{F343N} - m_{F435W}$ vs. $m_{F275W} - m_{F343N}$ two-color diagrams that we show for SGB and HB stars (middle panels) and the $\Delta_{CF275W,F343N,F435W}$ vs. $\Delta_{F275W,F814W}$ ChMs of RGB, AGB, and MS stars (bottom panels).

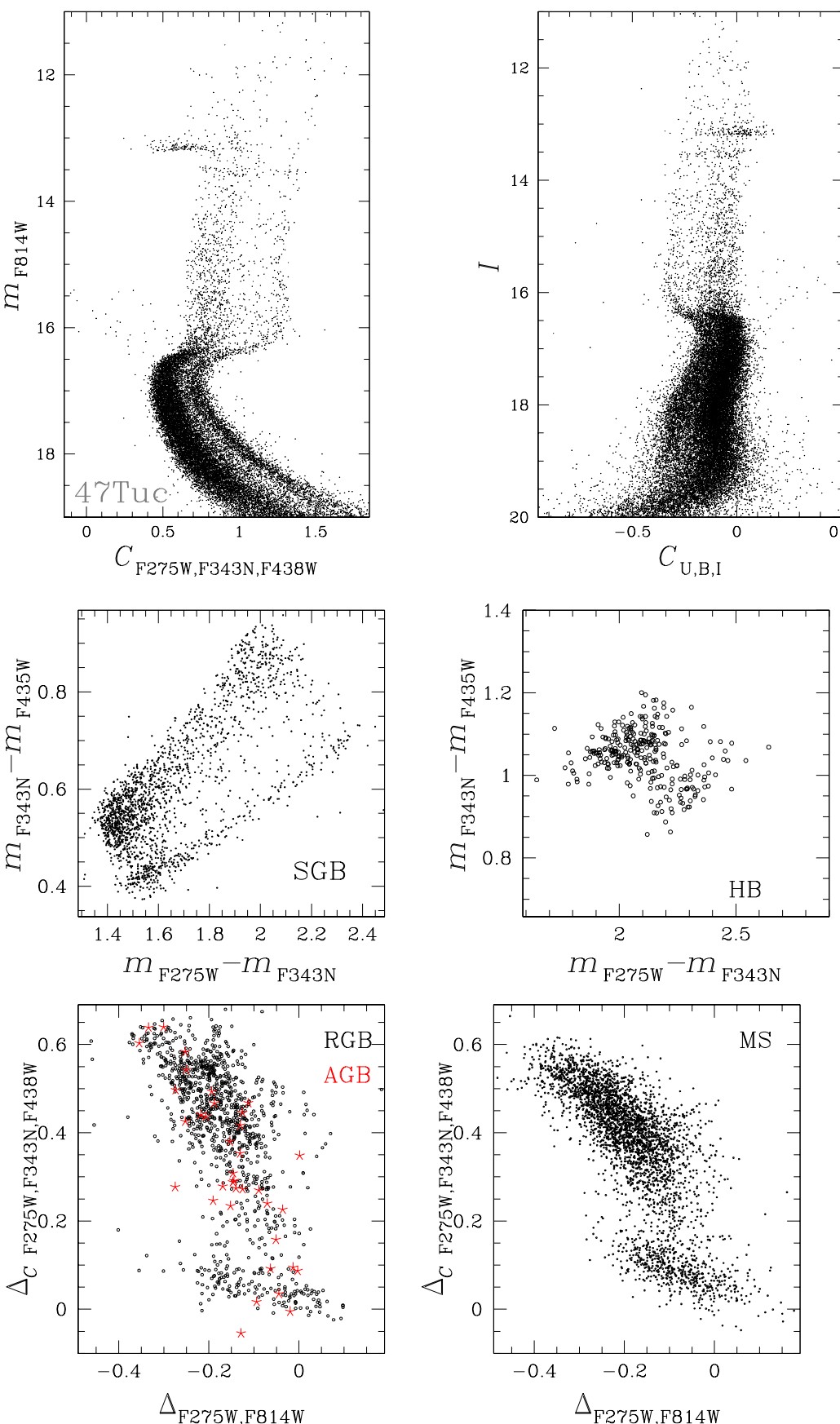

**Figure 1.** Collection of photometric diagrams to identify multiple populations in 47 Tucanae. The figures are derived from the photometric catalogs by Milone et al. [23,61].

As an example of methods to identify multiple populations among M dwarfs, we use the catalogs by Milone et al. [23,61] and Dondoglio et al. [33] to plot various photometric diagrams of 47 Tucanae in Figure 2. Specifically, we show the deep $m_{F814W}$ vs. $m_{F606W} - m_{F814W}$ CMD (panel a) and highlight the region of the MS where multiple populations are more evident (panel b). In these diagrams, the MS broadening is much wider than the spread due to observational errors alone, thus revealing the multiple populations. Hints of parallel sequences are also present. The $m_{F160W}$ vs. $m_{F110W} - m_{F160W}$ CMD is illustrated in panel d and reveals the effectiveness of NIR photometry to identify multiple populations among M dwarfs. While stars brighter than the knee[2] around $m_{F160W} \sim 20.2$ span a narrow color range, the MS breadth suddenly increases from the MS knee toward faint magnitudes. A gradient in the $m_{F110W} - m_{F160W}$ color distribution is also evident, with the majority of stars having blue colors and a tail of red MS stars. Finally, the $\Delta_{F110W,F160W}$ vs. $\Delta_{F606W,F814W}$ ChM of M dwarfs and the corresponding Hess diagram are plotted in panels c1 and c2 of Figure 2. The ChM reveals an extended 1P sequence composed of stars with $\Delta_{F110W,F160W} \lesssim 0.25$ and three main groups of 2P stars in close analogy with what is observed along the upper MS and the RGB.

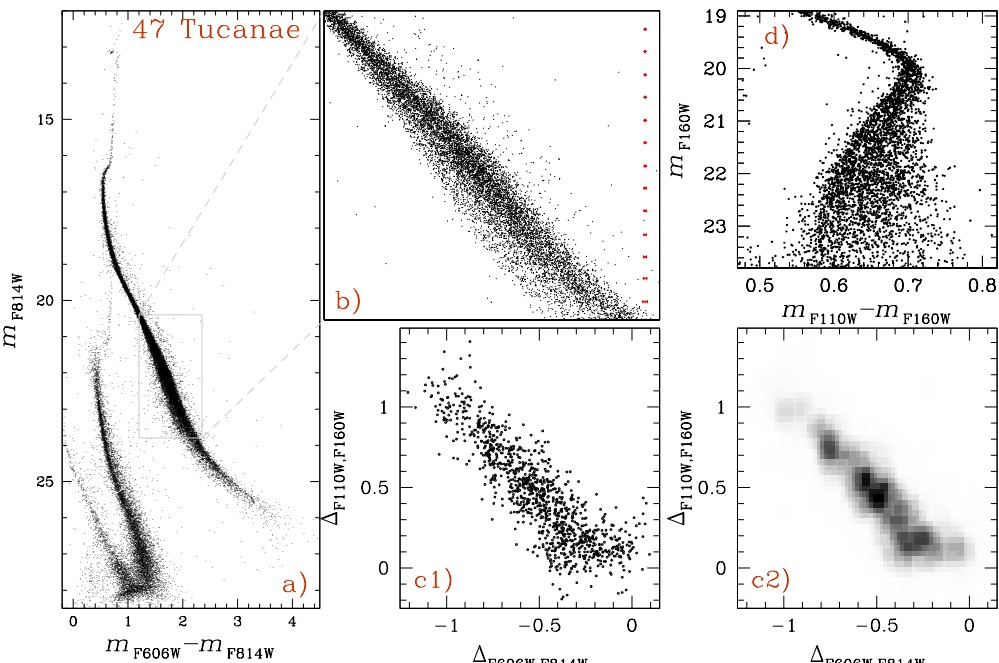

**Figure 2.** Collection of photometric diagrams that highlight the multiple populations among M dwarfs in 47 Tucanae. Panel (**a**) shows the optical $m_{F814W}$ vs. $m_{F606W} - m_{F814W}$ CMD, whereas panel (**b**) is a zoom around the MS region where multiple sequences are more evident. Panels (**c1,c2**) represent the ChM of M dwarfs and the corresponding Hess diagrams, respectively. The NIR $m_{F814W}$ vs. $m_{F606W} - m_{F814W}$ CMD is plotted in panel (**d**). To prepare this figure we used the photometric catalogs by Milone et al. [23,61] and Dondoglio et al. [33].

## 3. Classification of Globular Clusters

As we will widely discuss through this review, various properties of multiple populations, such as the the chemical compositions (Section 5), the fractions of 1P and 2P stars, the number of distinct sub-populations that compose the 1P and 2P (see Section 6), spatial distributions (Section 6.6), and kinematics (Section 6.7) dramatically change from one cluster to another. With this in mind, we identify two distinct classifications of GCs that we will use throughout the paper.

- GCs are grouped into two main groups of **Type I and Type II** based on the distribution of stars in the ChM and/or on star-to-star heavy-element variations [24,64]. Type I

GCs, which comprise the majority of Galactic GCs, host two main populations of 1P and 2P stars with similar metallicity but different abundances of some light elements. The second group of GCs, dubbed Type II GCs, comprises about 15–20% of the Milky Way GCs studied through the ChM. Type II GCs are characterized by at least one of these properties:

1.  Two or more sequences in the ChM. In addition to ChM composed of 1P and 2P stars, they show at least an additional sequence running on the red side of the main map [24].
2.  Split or broad SGBs in both UV and optical CMDs. Stars on the faint SGB evolve into a distinct RGB sequence with redder $U - I$ colors than the remaining RGB stars (hereafter red-RGB) [24,65,66].
3.  At odds with the majority of GCs, they exhibit significant variation in metallicity, due to iron variation, overall C+N+O, variations, or both (e.g., [20,28,67–75]).

These three characteristics could be physically connected to each other as observed in some Type II GCs. As an example of this classification, we compare in Figure 3 the ChMs of the Type I GC NGC 6723 and the Type II GC NGC 1851 [24].

- Another classification is based on the color distance between the RGB and the reddest part of the HB ([76], *L1*). By adopting the names of the prototype GCs, we define **M3-like** those GCs with $L_1 < 0.35$, while the remaining clusters are named **M13-like** GCs. The HB of M3-like is well populated on the red side of the RR Lyrae instability strip, whereas in M13-like GCs, stars redder than the RR Lyrae are very few [76,77]. Other differences between these two groups of GCs comprise the evidence that (i) their SGBs have different slopes [78] and (ii) M3-like GCs exhibit less extended 1G ChM sequences than M13-like GCs with similar metallicities [79]. As an example, Figure 4 compares the CMDs and the ChMs of the prototype GCs M3 and M13.

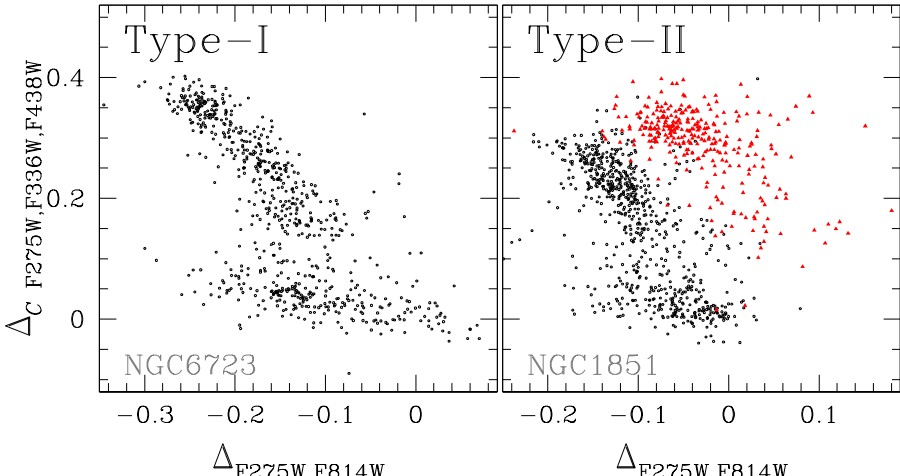

**Figure 3.** Chromosome maps of NGC 6723 (**left**) and NGC 1851 (**right**), which are prototypes of Type I and Type II GCs. Red symbols mark the red-RGB stars of NGC 1851. Photometry is from Milone et al. [24].

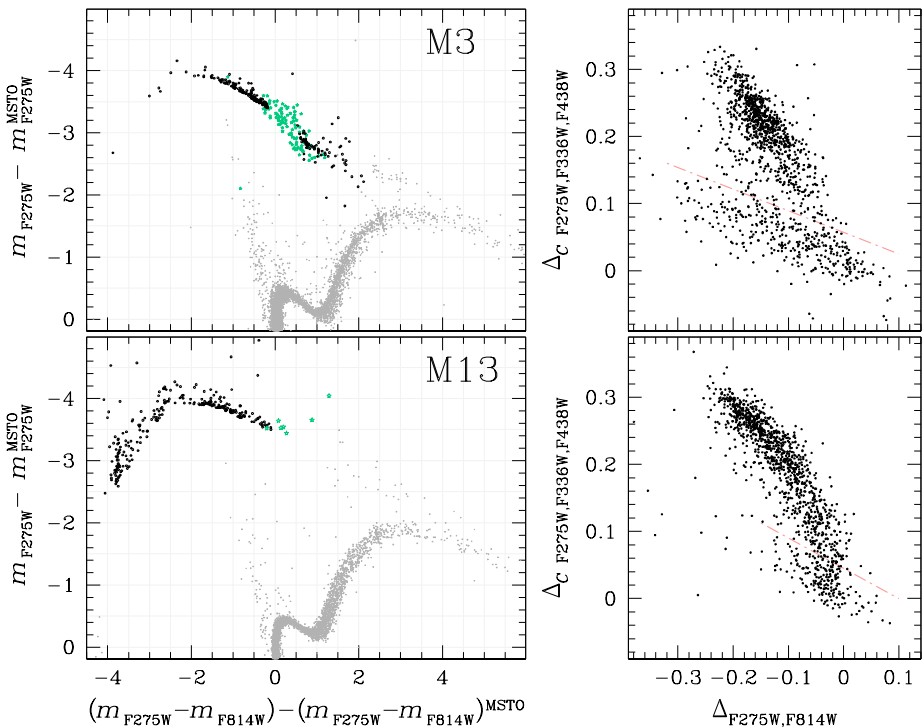

**Figure 4.** Comparison of the CMDs of M3 (**top left**) and M13 (**bottom left**). For comparison purposes, stellar colors and magnitudes are normalized to the corresponding values for the MS turn off. HB stars are represented with black dots, while variable stars are marked with aqua starred symbols. The ChMs of RGB stars of M3 and M13 are plotted in the top-right and bottom-right panel, respectively. The red dotted lines separate the bulk of 1P and 2P stars. This figure is derived by using photometry from Milone et al. [24].

## 4. Formation Scenarios

In the past ∼40 years, astronomers have developed many scenarios to explain the formation and evolution of multiple populations in GCs and proposed various candidate polluting stars to reproduce the chemical composition of 2P stars.

Historically, the so called Evolutionary Hypothesis suggested that light-element variations are the result of internal processes. Deep mixing of nuclear processed material from the interior into the surface layers would be the main force responsible for the abundance variations observed in RGB stars. However, this possibility has been ruled out by the discovery of star-to-star light-element variations among non-evolved MS stars which have negligible external convective zones [80,81]. The evidence of abundance variations of some light elements such as Na, Al, and Mg indicates that these elements are not forged in the small mass that we observe today but are likely produced in the interior of more massive stars [82]. Indeed, present-day stars are not hot enough to activate the Ne–Na and Mg–Al chains that are responsible for their production [83].

In the following, we summarize some of the most appealing scenarios, while in the next sections we present the main observational challenges. Most scenarios can be classified into two main groups. Multi-generation scenarios predict that GCs experience a prolonged star formation with the occurrence of multiple star-formation bursts. In these scenarios, second-generation stars would form from the gas polluted by more massive 1P stars. This gas can mix with some original material from which the 1P formed (e.g., [29–32,84–88]).

One of the most fascinating consequences of multi-generation scenarios is the so-called **mass-budget phenomenon**, which consists of the possibility that GCs were much more massive at formation by a factor of ∼5–20 (e.g., [89]). This is because only a small fraction of the initial mass of 1P stars is ejected with the composition required to generate 2P stars. To account for the the large amount of 2P stars observed in present-day GCs, the multi-

generation scenarios require that the progenitors of today's GCs were substantially more massive than the present-day GCs and that they have lost most of their initial 1P into the field. A consequence is that GCs would provide a significant contribution to the assembly of the Galactic halo and, to less extent, of the Bulge (e.g., [5,90], and references therein). Clearly, such massive proto-GCs may have contributed to the cosmic reionization [5,91,92].

Alternative scenarios suggest that all GC stars are coeval, and the peculiar chemical composition of 2P raises from accretion phenomena that occur during the pre-MS phase [34,35]. We summarize in the following some of the most popular scenario for the origin of multiple populations. We will emphasize the main observational challenges to these scenarios.

### 4.1. The Asymptotic Giant Branch Scenario

Since the 1980s, it has been suggested that the chemical anomalies in GCs arose from intermediate-mass (between $\sim$3–4 and $\sim$8$\mathcal{M}_\odot$) stars during their asymptotic giant branch (AGB) evolution [29,30,93]. These stars experience the so-called hot bottom burning process in which their surface convective envelope reaches deep enough that the temperature exceeds $\sim$30–40 MK. Such temperatures are hot enough to allow nuclear burning to take place, resulting in efficient $p$-capture nuclear processing.

The AGB scenario predicts that GCs experience a major star formation event that leads to the formation of 1P stars. The intra-cluster medium is then cleaned by the feedback from SNe explosions so that virtually all material enhanced in metals is lost by the proto-GC. When intermediate-age stars experience the AGB phase, their low-velocity ejecta ($\sim$10 km/s) are retained within the cluster potential well. Then, second-generation stars form in the cluster center, where the AGB ejecta together with pristine material accumulate via cooling-flow. Dilution with gas sharing the same chemical composition as 1P stars is a crucial ingredient for the AGB scenario, which allows for reproducing the O–Na anticorrelations observed in GCs.

In this scenario, the interaction with the Galaxy, together with the expansion of the cluster external regions associated with 1P SNe, is responsible for the loss of a large fraction of GC stars during the early phases of its formation. Early mass loss would mostly affect 1P stars, whereas the 2P stars, which are initially clustered in the GC center, are largely retained within the proto-cluster [31,87,94].

### 4.2. Fast-Rotating Massive Star Scenario

In this scenario, the products of hot hydrogen burning are generated in the core of massive stars that rotate near the break-up limit. The polluted material is brought up to the stellar surface as a result of rotation-induced mixing, which is responsible for a nearly full mixing of the gas in the star. These fast-rotating massive stars (FRMSs) lose mass through slow mechanical equatorial winds and eject the polluted material in their surrounding discs, where the formation of 2P stars occur. To better reproduce the observed chemical composition of 2P stars, the ashes of the FRMSs that form the 2P are diluted with the gas left over from the formation of 1P stars [86,95,96].

### 4.3. Massive Interacting Binaries

In de Mink et al. [84], the authors proposed massive binaries as sources of the polluted material from which the 2P stars are eventually formed. To investigate this possibility, they computed the evolution of a 20$\mathcal{M}_\odot$ star interacting with a 15$\mathcal{M}_\odot$ companion. This binary system sheds almost the entire envelope of the primary star, which corresponds to about 10 solar masses of processed material. Such stellar ejecta, which share the same abundance patterns of He, C, N, O, Na, and Al as observed in GCs, have slow velocities so that they remain within the potential well of the proto-GC. In this scenario, second stellar generations form from subsequent star-formation episodes involving polluted material diluted with pristine gas.

Massive interacting binaries, together with fast-rotating massive stars, are also responsible for the chemical composition of 2P stars in the **early disc accretion scenario**. In this scenario, the enriched material released by interacting massive binaries and FRMSs is accreted on the protoplanetary discs of pre-main sequence stars and ultimately on the young stars [34]. At odds with the original scenarios by de Mink et al. [84], Decressin et al. [96], this scenario does not imply multiple episodes of star formation.

### 4.4. Multiple Stellar Populations as a Case of Cooling

Recently, Renzini et al. [88] proposed that most GCs formed inside pre-galactic dwarfs, in an epoch when the main body of the Milky Way was not yet formed. The conditions for generating multiple populations also occurred during the Galactic Bulge formation in metal-rich GCs. However, such conditions were rarer in the Bulge than in the dwarfs.

Since 2P stars of most GCs share similar abundances of heavy elements as 1P stars, the material from which 2P stars form must not be polluted by the heavy elements ejected by supernovae. To avoid such kind of contamination, Renzini and collaborators suggested that the most massive stars would sink into black holes (see also [86]). Stars more massive than a certain threshold ($\sim$20–40$\mathcal{M}_\odot$) would avoid the supernovae phase, thus suppressing most of the star formation feedback shortly after the formation of the first generation GC stars and for a period of $\sim$5–10 Myr.

The main implication of suppressing star formation feedback is a runaway star formation, where the residual gas, together with the binary star ejecta, keeps forming stars until supernovae begin. Hence, the CNO/$p$-capture materials, which are ejected by massive binaries from previous generations during the common envelope phase, are responsible for the chemical composition of 2P stars. Intriguingly, this scenario is similar to the overcooling that, in the absence of feedback, would have turned all the baryons into stars in the early Universe [88].

### 4.5. The Super-Massive-Star Scenarios

Stars more massive than $10^3$ solar masses are considered by Denissenkov and Hartwick [32] as responsible for the pollution of the gas from which 2P stars formed. In this scenario, the massive cluster stars would first sink in the center as a result of dynamic friction and coalesce there, thus forming a super-massive star.

Main-sequence stars with these masses are convective, and their luminosity can exceed the Eddington limit. Hence, they would lose a significant amount of mass as a result of various instabilities and stellar winds. The mass lost by the super-massive stars, which is enriched in helium and in the products of CNO-cycling and $p$-capture reactions, then mixes with the original material, thus forming the second generation of stars.

Gieles et al. [35] proposed another scenario where super-massive stars are responsible for the chemical composition of 2P stars. Following the idea by Denissenkov and Hartwick, they suggest that due to gas accretion the proto-GC undergoes an adiabatic contraction and dramatically increases the star-collision rate. When the cluster reaches high density, this phenomenon leads to the formation of a super-massive star (SMS) via runaway collisions. Hence, at odds with the scenario by [32], Gieles and collaborators predict one star formation episode only, but some of the 1P stars are polluted by the super-massive star ejecta.

The polluted material released by the SMS winds is then diluted with pristine gas and accreted onto the protostars, thus forming 2P stars. One of the main advantages of this scenario is that the SMS can accrete a rate of mass in stars that is comparable with its mass loss. Because of this rejuvenation treatment, the total amount of material ejected by the SMS can be an order of magnitude higher than the SMS maximum mass.

### 4.6. Stellar Mergers

Mergers of massive stars in initially binary-rich embedded proto-GCs have been recently considered as possibly responsible for the occurrence of multiple stellar populations in GCs [97]. N-body modeling of very young clusters shows that a large fraction of more

than half of the massive stars ($\mathcal{M} > 30\mathcal{M}_\odot$) can merge within the first five million years of cluster life.

The polluted material ejected during the stellar merger can mix with the winds of the fast-rotating massive stars and supermassive stars that formed after mergers. In this scenario, the 2P stars form from residual embedding gas mixed with the gas polluted by merger-driven ejection/winds. Hence, multiple polluters contribute to the chemical composition of 2P stars.

## 5. The Chemical Composition of Multiple Populations

GC stars include two main groups of 1P and 2P stars with different chemical compositions. 1P stars are indistinguishable from the field, as they share similar chemical composition as field stars with the same metallicity. The 2P stars are mostly observed GCs. They exhibit peculiar light-element abundances, which are uncommon for field stars. We summarize in this section the chemical properties of stellar populations in GCs through the abundances of those elements that better characterize the multiple stellar population phenomenon in GCs.

### 5.1. Helium

Although helium is the second most abundant element in stars, spectroscopic estimates of helium abundances in GC stars are quite rare. The main challenge is that reliable helium estimates from photospheric spectral lines are only feasible for HB stars with effective temperatures between $\sim$8000 K and $\sim$11,500 K [98,99]. Colder stars do not have strong enough helium lines in the optical, while stars hotter than $\sim$11,500 K are affected by helium settling. Hence, the helium content of their atmospheres is not representative of the total helium abundance [100–102].

Direct spectroscopic evidence of helium-enhanced stars in the GCs is provided by NLTE analysis of the photospheric helium lines at 5875.6 Å in 17 blue HB stars of NGC 2808 colder than 11,500 K. The results showed that stars in the studied HB segment have average helium content of Y$\sim$0.34. Hence, they are enhanced in helium-mass fraction by $\Delta$Y$\sim$0.09 with respect to the primordial value [99]. Evidence of extreme helium enhancement by more than $\Delta$Y$\sim$0.15 is provided by the investigation of NIR chromospheric He lines in RGB stars of both NGC 2808 and $\omega$ Centauri [103,104].

Recently, multiband photometry has provided a major breakthrough in constraining the helium abundances of multiple populations in GCs. This method is based on the comparison of the observed colors of the distinct populations and colors derived from grids of synthetic spectra with appropriate chemical compositions [22,53]. The magnitude differences of the RGB bumps of 1P and 2P stars from multiband photometry are powerful tools to constrain their relative helium contents [56,105]. Helium estimates are now available from homogeneous analysis of more than 70 GCs in the Milky Way and in the Magellanic Clouds [61,105–107].

The helium difference between 2P and 1P stars is, on average, $\Delta Y_{2P-1P}\sim$0.01 in helium-mass fraction. The maximum internal helium variations range from less than $\Delta Y_{max}\sim$0.01 to 0.18, with NGC 2419 being the GC with the largest internal helium variation [61,105,108].

The helium content of 2P stars is a constraint for multiple population formation scenarios. The fast-rotating massive star scenario predicts that 2P stars have helium abundances between that of 1P stars and $\sim$0.8, whereas the maximum helium enhancement expected in the AGB scenario is about $\sim$0.36–0.38 [109–111].

Helium variations strongly correlate with the mass of the host GC, with massive clusters typically hosting stars with extreme helium content [61,105]. In addition, the helium content of multiple populations correlates with their abundances of N, Na, and Al and anticorrelate with C, O, and Mg [56,64].

*5.2. Lithium*

Results based on high-resolution spectroscopy reveal that 1P and 2P stars of some GCs, (M4, M12 and NGC 362) share the same lithium abundance [112–114], while some 2P stars of other clusters such as M5, NGC 1904, NGC 2808, NGC 6397, and NGC 6752 have lower lithium abundances than the 1P[3]. However, significant lithium depletion is only associated with a minority fraction of 2P stars with extreme chemical composition [113,114,116–118]. As a remarkable example, Li-poor stars of NGC 2808 comprise only those 2P stars with the highest helium content (Y ∼ 0.36), whereas the remaining 2P stars, including 2P stars with Y ∼ 0.32, have the same lithium content as the 1P [64,114,119,120].

Since lithium is characterized by low burning temperature ($\sim 2.5 \times 10^6$ K), it would constrain the source of polluted material from which 2P stars formed. Massive polluters, such as massive binaries, supermassive stars, and fast-rotating massive stars, can only destroy lithium. In contrast, intermediate-mass AGB stars can activate the Cameron–Fowler mechanism during the hot bottom burning and produce lithium [121][4]. Hence, the evidence that 2P stars either share the same lithium abundance as 1P stars or exhibit moderate lithium depletion supports the AGB scenario.

*5.3. Light Elements*

One of the main features of multiple populations is that 2P stars exhibit different content of some light elements compared to 1P stars. Similar elemental variations are observed among RGB stars and unevolved MS stars, thus indicating that such chemical variations are not due to stellar evolution. The most studied elements include:

- **Carbon, Nitrogen, Oxygen, and Sodium.** 2P stars are enhanced in N and Na and depleted in C and O compared to 1P stars. These elements depict well-defined patterns such as the Na–O and C–N anticorrelations and Na–N and C–O correlations, which are ubiquitous features of GCs with multiple populations [6,122][5].
- **Magnesium, Aluminum, and Silicon.** Significant Mg and Si variations are present in a restricted number of massive GCs only, and the spread in [Mg/Fe] is typically wider in low-metallicity clusters. In contrast, aluminum variations are observed in nearly all GCs with [Al/Fe] correlating with [Na/Fe] (e.g., [128,129]). GCs with internal variations in these elements exhibit an Mg–Al anticorrelation and an Si–Al correlation. In these clusters, 2P stars are depleted in Mg and enhanced in Al and Si with respect to the 1P.
- **Potassium.** Internal variations in [K/Fe] have been detected in-only three massive GCs , namely NGC 2419, NGC 2808, and $\omega$ Centauri [130–133].

As an example, we show in Figure 5 some correlations involving C, N, O, Mg, Al, and Si for NGC 2808.

The distribution of light elements among GC stars and the maximum internal variation significantly change from one cluster to another. Some clusters exhibit bimodal or discrete stellar distribution in the Na–O plane (e.g., M4 and NGC 6752, [19,44]), whereas the distribution of stars in other GCs seems continuous [128]. Although such continuity seems in contrast with the discrete distribution of 1P and 2P stars along the ChM, the 1P stars identified on the ChM are O-rich and Na-poor, and 2P stars are oxygen-depleted and sodium enhanced.

Since the various light elements are formed and destroyed at different temperatures, the chemical patterns summarized above provide information on the sites where these processes occurred.

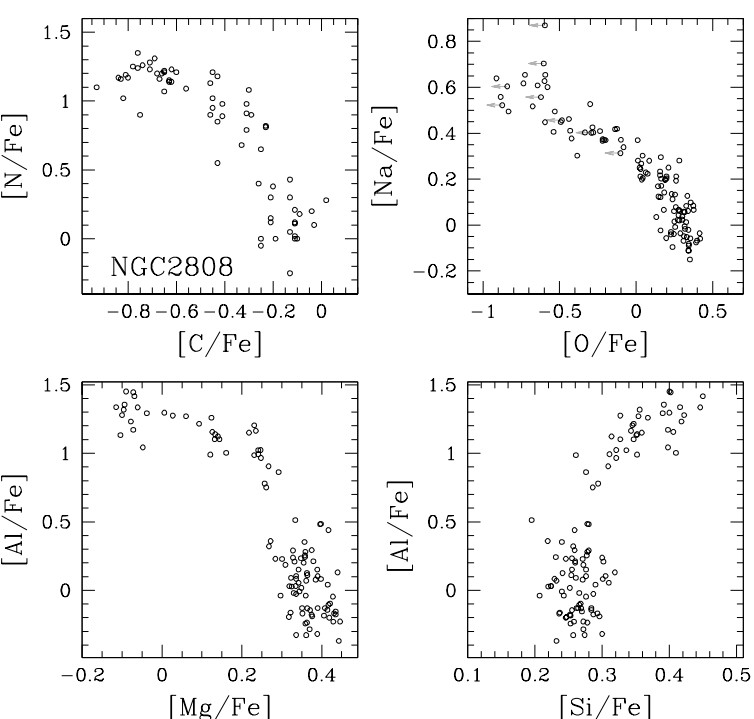

**Figure 5.** This figure shows the C–N, Na–O, Mg–Al anticorrelations and the Al–Si correlation for NGC 2808. C and N abundances are from Carlos et al. [134], whereas the remaining elements are taken from Carretta [119] and Carretta et al. [135].

The chemical patterns involving C, N, O, and Na may reflect the chemical composition that results from CNO-cycling and *p*-capture processes at high temperatures. In main-sequence stars, the conversion of carbon into nitrogen requires temperatures hotter than $\sim 10 \times 10^6$ K. The activation of the ON branch of the CNO cycle and the Ne–Na chain, which produces Na from Ne and is responsible for the Na–O anticorrelation, requires higher temperatures of more than $\sim 35 \times 10^6$ K [136]. In contrast, the abundance ratio of Mg and Al would be associated with the nuclear reactions that occurs at high temperatures such as the $^{24}$Mg (p$\gamma$)-Al$^{25}$, which requires about $\sim 70 \times 10^6$ K (e.g., [90,95]). Temperatures hotter than $\sim 80 \times 10^6$ K are needed to produce silicon, whereas potassium variations are due to the Ar–K chain that occurs at extreme temperatures of more than $\sim 150 \times 10^6$ K [83,137].

Given that stars of different masses reach different temperatures in their interiors, the chemical composition of 2P stars would constrain the nature of the stars that polluted the material from which they formed. As an example, the maximum temperature in fast-rotating massive stars and massive binaries would not exceed $\sim 65 \times 10^6$ K, thus challenging the possibility that they are the only sources responsible for the chemical composition of Mg-depleted stellar populations [32,87]. The hot-bottom burning of AGB stars can span a wide range of temperatures up to more than $100 \times 10^6$ K, thus allowing various processes, including O depletion and Na production and Al, Si, and K production. A challenge for the AGB scenario arises from the massive AGB stars, which would destroy Na, thus producing a correlation between Na and O instead of the observed anticorrelation.

However, uncertainties on the cross sections of the reactions responsible for forging or destroying the various elements and the poor knowledge on the hot-bottom burning prevent us from firm conclusions (see discussion by [87,90]).

### 5.4. Metallicity Variations in Globular Clusters

Metallicity variations are commonly associated with stellar systems more massive than present-day GCs. For a long time, the homogeneity in metallicity has been considered one of the most distinctive traits of GCs (with the sole exception of $\omega$ Centauri). Indeed,

more massive systems, such as galaxies, have deeper potential wells required to retain the fast ejecta from Supernovae [6,127,138].

Thanks to studies based on high-resolution spectroscopy and on the ChMs, in the past decade, we have realized that internal variations in the overall metallicity are not a peculiarity of $\omega$ Centauri. Recently, the number of observations suggesting that metallicity spreads are quite a common phenomenon in GCs is becoming increasingly larger [61,66,68,69,71,72,75,139,140]. In this section, we discuss the two main phenomena that are as of now associated with differences in overall metallicity.

5.4.1. The Chemical Inhomogeneity of 1P Stars

One of the most exciting outcomes of the ChM analysis is the fact that the 1P sequence of the ChM of most GCs is intrinsically broad, and in some clusters, such as NGC 2808, it exhibits hints of bimodality [24,56]. The same phenomenon is observed for red HB stars, where the sequence of 1P stars in the $m_{F336W} - m_{F438W}$ vs. $m_{F275W} - m_{F336W}$ two-color diagram is not consistent with a chemically homogeneous stellar population [141]. The extended 1P sequence is not a peculiarity of RGB and red HB stars. The evidence of elongated 1P sequence among non-evolved MS stars is proof that the material from which 1P stars formed was not chemically homogeneous [79].

The extended 1P sequence was tentatively associated with pure helium variations [56,61] without any significant variation in C, N, and O. However, such a conclusion is at odds with expectations from standard nucleosynthesis (see discussion by [61]). It would imply that helium inhomogeneities are the product of exotic phenomena that occurred in the early Universe [142].

When high-resolution spectroscopy came into the picture, it revealed that iron variations within the material that originated the 1P can be responsible for the extended 1P sequence of the ChM [59,64]. Marino et al. [59] derived high-precision iron abundance determinations of 18 1P stars of NGC 3201 and detected a [Fe/H] spread of about 0.1 dex, with the iron abundance correlating with the $\Delta_{F275W,F814W}$ pseudo-color of 1P stars. A similar conclusion that metallicity spread is responsible for the extended 1P sequences in the ChM in the GCs NGC 6362 and NGC 6838 comes from the distribution of SGB stars in the appropriate In pseudo-two-magnitude diagrams [79].

It is possible to take advantage of the 1P F275W−F814W color extension of the RGB [24] to constrain the internal iron spread by assuming that metallicity spread is the only factor responsible for the color width of 1P stars. The internal variations in [Fe/H] range from less than ~0.05 to 0.30 dex and mildly correlate with the mass and the metallicity of the host GC. The 1P metallicity variation mildly correlates with cluster metallicity. For a fixed metallicity, 1P stars in M13-like GCs exhibit smaller iron variations than those of M3-like GCs [79].

The evidence of metallicity variations within the interstellar medium from which proto-clusters were born would constrain the multiple population formation scenarios. The metallicity scatter among 1P stars provides a lower-limit to the the scatter initially present in the gas [143–145]. Turbulent diffusion within a proto-cluster cloud should smooth out chemical inhomogeneities at the scale of the cloud in roughly a crossing time. Since the crossing time increases in low-density extended clouds, while decreasing in high-density turbulent clouds, the 1P metallicity scatter is indicative of both metallicity variation in the original gas and the diffusion efficiency during the cloud collapse [79]. In this context, 1P stars of M13-like GCs would form in a denser environment than 1P stars of M3-like GCs.

Another intriguing feature of the ChMs of NGC 6362 and NGC 6838, which are quite simple GCs in the context of multiple populations, is the fact that the $\Delta_{F275W,F814W}$ extensions of their 2P stars are narrower than those of 1P stars [79]. Hence, the material from which the 2P originated has a more homogeneous iron content than the cloud that formed 1P stars. Qualitatively, these observations are consistent with a scenario where the 1P formed in a low-density cloud, where the high crossing time has reduced the mixing efficiency, whereas the 2P originated in a high-density environment, as suggested by

multi-generation scenarios. Clearly, this finding seriously challenges the scenarios for the formation of multiple populations that are based on accretion of material processed in massive 1P stars onto existing protostars [79].

Further information on the origin of 2P stars would come from the relative iron contents of the distinct stellar populations. The pioneering paper by Yong et al. [146], based on high-precision differential abundances of multiple stellar populations in NGC 6752, first detected star-to-star metallicity variations, with the iron abundance increasing for 2P stars. The ChM analysis now has the potential of providing similar information for many clusters. As an example, Legnardi et al. [79] pointed out that the relative separation between 1P and 2P sequences in the ChM changes from one cluster to another (see Figures 3–7 from [24]). In some clusters (e.g., NGC 2808 and NGC 6981), the 2P sequences merge with the 1P at low values of $\Delta_{F275W,F814W}$; the 2P stars of other clusters such as NGC 104 and NGC 5272 are distributed on the metal-intermediate and possibly the metal-rich side of the 1P sequence. If the $\Delta_{F275W,F814W}$ value where 2P stars join the 1P is indicative of their relative iron abundance alone, 2P stars can be either enhanced or depleted in iron with respect to the 1P.

### 5.4.2. Chemical Composition of Type II, 'Anomalous' GCs

A more complex phenomenon associated with internal variations in metallicity is the presence of a genuine new class of 'anomalous' GCs with variations in heavy elements among 2P stars. From a chemical perspective, these clusters constitute a well-defined class of 'anomalous' GCs, with distinct photometric and spectroscopic properties than the other GCs. The discovery of anomalous GCs revealed how $\omega$ Centauri is not unique in harboring stellar populations with different overall metal content. Although this massive GC has the most extreme variations in the heavy elements ever observed in this group, it perfectly fits the qualitative chemical pattern characterizing this class of globulars.

A typical feature of most Type II GCs is the over-enrichment in the elements produced via slow neutron capture reactions (*s*-elements, [*s*/Fe]) in the stars also enhanced in metallicity [20,66,68,69]. The degree of *s*-elements enrichment varies from cluster to cluster. As an example, M22 shows a more moderate variation in all the *s*-elements compared to other Type II GCs with similar metallicity, e.g., M2 and NGC 5286. However, there is no evidence of *s*-processed-based enrichment in the two less massive Type II GCs, NGC 6934 and NGC 1261, suggesting that enrichment in the *s*-elements is not a *universal* feature of these objects and that the metals and *s*-elements enrichment are not coupled with each other [70].

The variations in [Fe/H] and [*s*/Fe] are very likely due to polluters with different mass ranges, i.e., to high mass and low mass stars, respectively. A possibility is that in more massive proto-clusters, the star formation proceeded for longer times than in normal clusters, giving the possibility to low-mass AGB stars to contribute to the enrichment in *s*-elements of the proto-cluster. At the time these low-mass stars start to pollute the intra-cluster medium, material enriched from fast SNe, and previously expelled, may have had the time to fall back into the cluster [70]. A similar scenario has been proposed by D'Antona et al. [147] for $\omega$ Centauri. If not a coincidence, the fact that the two less massive Type II GCs do not show evidence for *s*-element enrichment may suggest that the latest star-formation event occurred before the low-mass AGBs had the time to pollute the intra-cluster medium, pointing toward a less-extended star-formation history [70].

From a chemical perspective, this class of stellar systems sits in the middle between a genuine GC, the typical Type I GCs, and more massive systems such as dwarf galaxies whereby they share the enrichment in metallicity. An example of the *hybrid* nature of these objects is displayed in Figure 6, where we show chemical abundances for the Type II GC M22. The two groups of stars with different abundances of iron and *s*-process elements are selected from the left-panel plot, whereas the right panel clearly shows the Na–O anticorrelation, which is typically observed in GCs. A puzzling feature characterising several Type II GCs, including M2, M22, M54, $\omega$ Centauri, NGC 1851, NGC 5286, NGC 6273,

and NGC 6934 is that light-element variations and Na–O anticorrelations are present in stars with different metallicity (e.g., [28,59,66–69,73–75,148,149]). This fact seriously challenges our understanding of the chemical evolution of these objects.

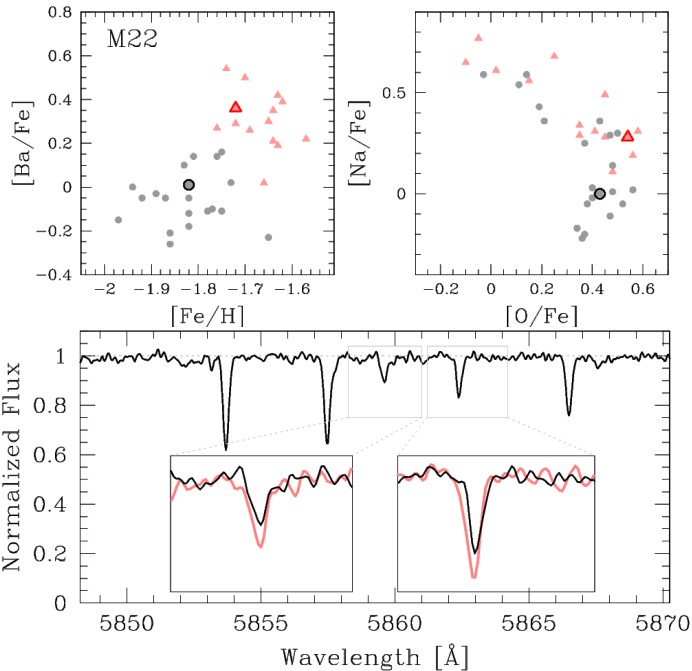

**Figure 6.** (**Upper panels**) Barium against iron (left) and sodium against oxygen abundance (right) for stars in the Type II GC, M22; *s*-rich/Fe-rich and *s*-poor/Fe-poor stars are represented with black and red symbols, respectively. (**Bottom panels**). Normalized spectrum of the star that is marked with large black circles in the upper panels. The black and red lines in the insets compare the spectra of the metal-poor and metal-rich star, respectively, (large symbols in the upper panels) centered around two FeI lines. Since these two stars share very similar stellar parameters, the fact that Fe lines differ significantly is a signature of metallicity difference. The spectra and chemical abundance that we use to draw this Figure are taken from [28,69].

Other well-studied Type II GCs include:

- **M54**, which is associated with the Sagittarius dwarf spheroidal. The fact that this cluster is located in the nucleus of a dwarf galaxy and belongs to the class of Type II GCs corroborates the idea that Type II GCs are naked nuclei of dwarf galaxies. Indeed, when the Sagittarius will be entirely stripped off by tidal interactions with the Milky Way, M54 will be deprived of its surrounding galaxy and will be almost indistinguishable from the other Type II GCs.

- The most massive Galactic GC, $\omega$ **Centauri**, exhibits large variations in metallicity, *s*-process elements, and overall C+N+O content [138,148–150]. Due to its extreme chemical composition, and retrograde orbit, $\omega$ Centauri is considered by many to be the surviving nucleus of a dwarf galaxy [151], an hypothesis reinforced by the recent discovery of tidal debris [152].

- Interestingly, the Type II GCs **M2** and **NGC 1851** are surrounded by extended envelopes [153–155], while Marino et al. [156] noticed that the chemical composition of stars in the stellar halo around NGC 1851 is compatible with a dwarf remnant.

- **Terzan 5** is the most metal-rich GC with metallicity variations. It hosts at least two main stellar populations with [Fe/H] $\sim$ $-0.2$ and [Fe/H] $\sim$ $+0.3$. Together with **NGC 6388**, it is a Type II GC located in the Galactic Bulge. Due to its metal content, it has been suggested that it is the remnant of a building block of the Bulge [157–159].

- The metal-poor GC **M15** ([Fe/H] $\sim$ $-2.4$) hosts two groups of stars with the same iron abundance but different content of barium and europium, in contrast to most GCs that

have constant [Eu/Fe]. Hence, the nucleosynthetic history seems dominated more by the *r*-process than material in solar system [160]. Each group of *r*-poor and *r*-rich stars exhibit a distinct Na–O anticorrelation as observed for *s*-rich and *s*-poor stars of M22 and other Type II GCs. The red population of the ChM of M15 hosts the ∼5% only of the total number of stars [161], whereas the two groups of *r*-rich and *r*-poor stars have comparable numbers of stars. As a consequence, there is no correspondence between red RGB stars of the ChM and stars with different content of *r*-elements.

## 6. The Properties of Multiple Populations

Results by several authors have vividly revealed the complexity of the multiple population phenomenon, in terms of the number of stellar populations, their chemical properties, and cluster-to-cluster heterogeneity. A major step forward in depicting the main properties of the multiple populations comes from the analysis of a homogeneous sample of ChMs for 60 GCs.

- **A widespread phenomenon.** The recent surveys of GCs have revealed that multiple populations are present in most studied Galactic GCs [24,25].
  Remarkable examples of simple population Galactic GCs comprise Ruprecht 106 and Terzan 7, which according to both spectroscopic studies and multiband photometry are consistent with simple populations [162–164]. Additional candidate simple population GCs comprise AM1, Eridanus, Palomar 3, Palomar 4, Palomar 14, and Pyxis, as inferred from their HB morphology [76,165].

- **GC specificity.** While 2P stars are present in nearly all GCs, they are rare in the Milky Way field, where they constitute only a few percent (∼1–3 %) of the field stars in the Galactic Halo (e.g., [166,167]) and in the inner Galaxy [168]. However, recent work suggests that at 1.5 kpc from the Milky Way center, 2P stars include $16.8^{+10.0}_{-7.0}$% of the total halo mass. The fraction drops to $2.7^{+1.0}_{-0.8}$% at 10 kpc [169]. These stars are generally thought to be either former members of dissolved GCs or stars of existing GCs that are lost into the field through interactions with the Milky Way (e.g., [170]). As an alternative, it has been also speculated that 2P stars are not a distinctive feature of GCs but also form in other stellar systems (e.g., [168]).

- **Variety.** GCs host different numbers of stellar populations, and the extension and morphology of the ChM changes from one cluster to another [24,106]. Moreover, the number of distinct sub-populations ranges from 2 (as in NGC 6535) to more than 17 in $\omega$ Centauri [24,171]. The collection of ChMs plotted in Figure 7 highlights the variety of the multiple population phenomenon.

- **1P-2P discreteness.** The ChMs of most GCs are composed of two main clumps of 1P and 2P stars. Although a few stars often populate the region of the ChM between the clumps, 1P and 2P stars are typically associated with discrete stellar populations in contrast to a continuous stellar distribution. The distribution of 2P stars in the ChM changes from one cluster to another. In some GCs, such as NGC 2808, we observe distinct stellar clumps, whereas 2P stars of other clusters (e.g., NGC 5272) exhibit nearly continuous pseudo-color distributions. In contrast, stellar clumps among 1P stars are quite rare, with NGC 2808 being a possible exception.

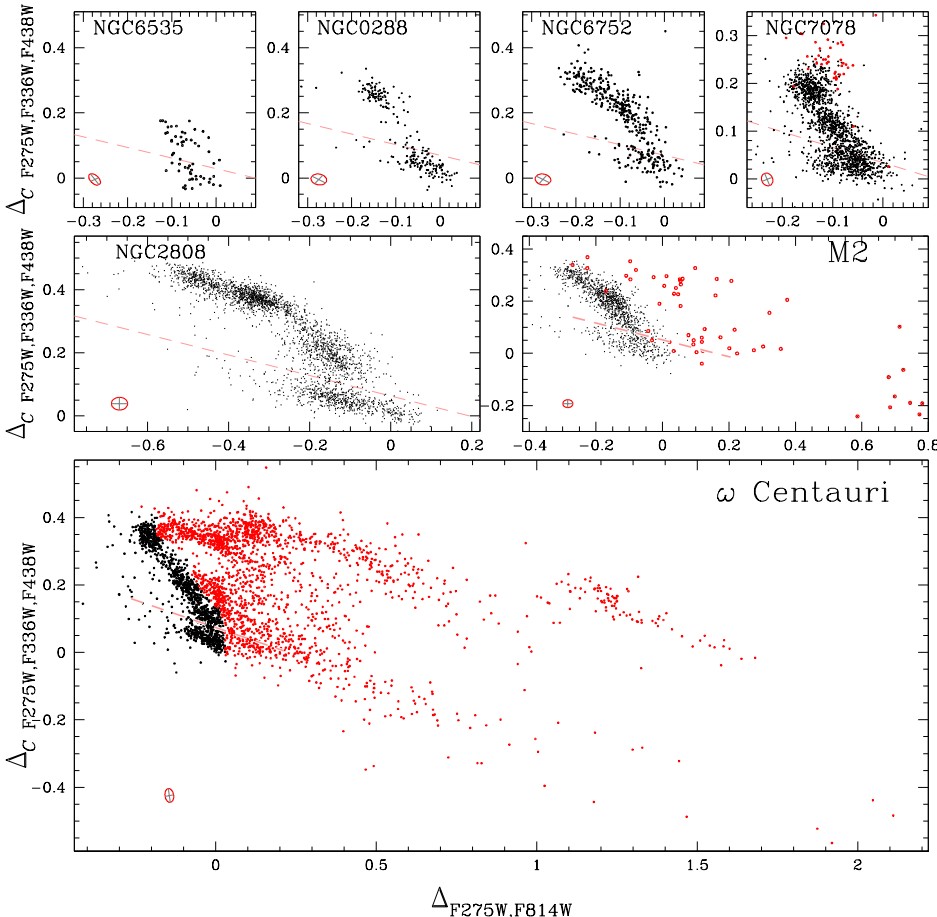

**Figure 7.** Chromosome maps for seven GCs. The dashed lines separate the bulk of 1P stars from the 2P. Red RGB stars are colored red. This figure illustrates the high degree of variety of the multiple population phenomenon.

Although the multiple population phenomenon is characterized by a high degree of variety, multiple stellar populations share several common properties. In the following, we summarize the main relations between multiple populations and the properties of the host cluster and the parent galaxy. Moreover, we discuss the relation with stellar mass.

### 6.1. Dependence on Cluster Mass

Recent work has shown that the complexity of the multiple population phenomenon increases with the mass of the host GC. The dependence on cluster mass involves the relative numbers of 1P and 2P stars as well as the internal chemical variations.

- In Milky Way GCs, the fraction of 2P stars ranges from less than 40%, in low-mass clusters such as NGC 6362 to more than 90% in the most massive GC, $\omega$ Centauri. As shown in the left panel of Figure 8, the fraction of 1P stars significantly anticorrelates with present-day cluster mass, and the significance of the anticorrelation increases when initial mass estimates are considered [24,106,141]. Additional relations involving the fraction of 1P stars comprise (i) a strong anticorrelation with the present-day mass of the 2P but (ii) a mild anticorrelation with the present-day mass of 1P stars [106].
- The internal variations in some light elements also depend on cluster mass (right panels of Figure 8). As an example, the $C_{F275W,F336W,F438W}$ and $C_{F336W,F438W,F814W}$ RGB widths, which are proxies of nitrogen abundance, correlate with cluster mass [24,164], in close analogy with what is observed for the $m_{F110W} - m_{F160W}$ color width of the MS below the knee [33], which is indicative of oxygen internal variations. Similarly,

the maximum internal variation of helium strongly correlates with cluster mass (right panel of Figure 8, [61]).

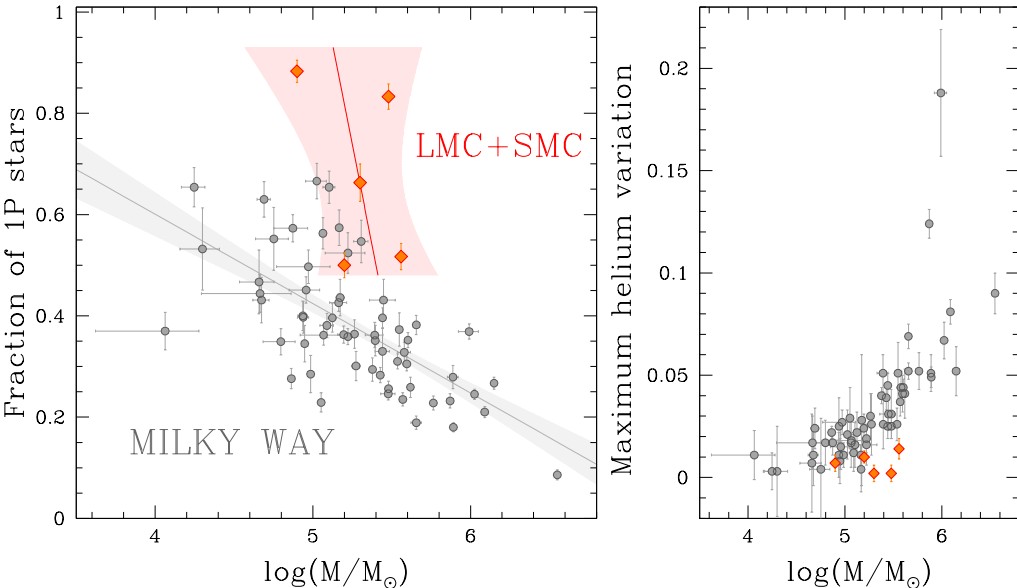

**Figure 8.** Fraction of 1P stars (**left**) and maximum internal mass-fraction helium variation (**right**) as a function of the present-day mass of the host GC. Gray and orange symbols represent Galactic and Magellanic-Cloud GCs, respectively. The gray and orange lines plotted in the left panels are the best-fit straight lines for the groups of GC with the same color. Masses are from Baumgardt and Hilker [172]; fraction of 1P stars and helium abundances are taken from Milone et al. [24,106], Zennaro et al. [108], Dondoglio et al. [141], D'Antona et al. [173] and from Milone et al. [61], Zennaro et al. [108], Lagioia et al. [164], and D'Antona et al. [173], respectively.

Type II GCs are among the most massive GCs of the Galaxy, thus suggesting that high cluster mass is required to generate their metal-rich stellar populations [66,174]. Clusters with metallicity variations are commonly associated with more massive stellar systems such as dwarf galaxies, which have been able to retain the fast ejecta from SNe. This idea is supported by the Type II GC M54, which is located in the nucleus of the Sagittarius dwarf galaxy. As illustrated in Figure 9, the role of the total mass in the evolution of this class of objects is corroborated by the correlation between the present-day mass and the additional iron locked in the stellar populations enhanced in metals. The latter corresponds to the difference in iron mass fraction between red RGB and blue RGB stars multiplied by the fraction of mass in the stellar populations associated with the red RGB [70,175]. Interestingly, low-mass Type II GCs exhibit homogeneous content of *s*-process elements, thus suggesting a mass threshold for the occurrence of the s-enrichment.

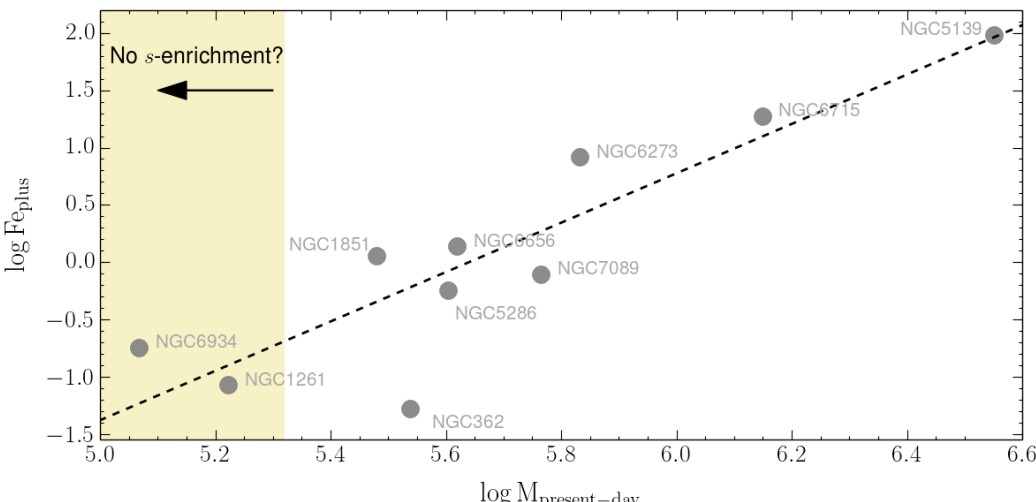

**Figure 9.** Logarithm of the additional iron included in the metal richer stellar populations as a function of the logarithm present-day GC masses.

### 6.2. Dependence on Cluster Orbit

Despite the strong correlation between the fraction of 1P stars and cluster mass, it is worth noticing second-order dependence with the GC orbit. Galactic GCs with large perigalactic radii host, on average, larger fractions of 1P stars than clusters with small perigalactic radii and similar present-day masses [106,108]. In addition, as shown in the left panel of Figure 8, the LMC and SMC clusters host larger fractions of 1P stars than Milky Way GCs with similar masses [106].

The dependence on cluster orbit [106,108] can be combined with the four relations discussed in Section 6.1 to constrain the evolution of multiple populations. Specifically, we consider: (i) the mild anticorrelation between the fraction of 1P stars and the total present-day mass of 1P stars, together with the anticorrelations between the fraction of 1P stars and (ii) the present-day GC mass, (iii) the initial GC mass, and (iv) the present-day total mass of 2P stars [106]. These relations may represent the smoking gun that GCs preferentially lose 1P stars. Indeed, they are expected in a scenario where the fraction of 1P stars ranges from ∼0.6 to 0.8 at cluster formation and decreases with increasing cluster mass. In this scenario, the GCs have lost a large fraction of their 1P stars but a smaller amount of 2P stars [106].

### 6.3. Multiple Populations and Stellar Mass

Very-low mass (VLM) stars of GCs are still poorly investigated in the context of multiple populations as they require high-precision NIR photometry of faint stars. Nevertheless, the comparison of the multiple population properties among stars of different masses may allow us to discriminate among the different formation scenarios. Pioneering work, based on *HST* photometry in the F110W and F160W bands of NIR/WFC3, provided the first results on the mass functions (MFs) and the chemical compositions of VLM stars.

- Deep *HST* photometry in NIR bands allowed disentangling the three distinct stellar populations of NGC 6752 below the MS knee and constraining their oxygen abundances. The discovery that the relative oxygen values of multiple populations among VLM stars and more massive ($\sim 0.8 \mathcal{M}_\odot$) RGB stars are consistent with each other suggests that the chemical composition of multiple populations does not depend on stellar mass [176]. A similar conclusion is provided for 1P and 2P stars of M4 [177].
- Dondoglio et al. [33] first derived the MFs of the multiple populations of NGC 2808 and M4 over the mass interval between $\sim 0.25$ and $0.80 \mathcal{M}_\odot$. The conclusion is that 1P and 2P stars share similar MFs and that the relative numbers of 1P and 2P stars are constant over the analyzed mass interval. Similar results have been found for

NGC 6752, where the fractions of stars in the three populations do not change with cluster mass [176].

Results on the oxygen variations among stars of different masses would constrain the scenarios for the formation of multiple populations. Indeed, if accretion of polluted material onto pre-existing stars is the mechanism responsible for the origin of 2P stars [34,84], the chemical composition of 2P stars could depend on stellar mass. As an example, by assuming Bondy–Hoyle–Littleton accretion, the amount of accreted material is proportional to the square of the stellar mass. Hence, VLM stars would accrete a smaller amount of polluted gas and exhibit smaller oxygen variations than RGB and bright MS stars.

The evidence that stellar populations exhibit the same present-day MFs is consistent with a scenario where they originated with similar initial mass functions (IMFs) [170]. If the distinct stellar populations formed in environments with different densities, such that 2P stars originated in the dense and compact regions of the cluster center (e.g., [94,178]), results on NGC 6752, M4 and NGC 2808 may indicate that the IMF does not significantly depend on the density in the formation environment [33].

### 6.4. Multiple Populations along the Asymptotic Giant Branch

Photometric surveys of multiple populations reveal that the fraction of 1P and 2P stars along the AGB of several GCs is comparable with that observed along the RGB and the MS [26], thus confirming previous results based on both photometry and spectroscopy [56,179–183]. In contrast, the AGB sequences of at least ∼25% of the studied GCs host lower fractions of 2P stars than the RGBs [164]. The most studied cases include NGC 6752, NGC 6266, and NGC 2808 [26,179,184–186], where only the 2P stars with the most extreme chemical composition seem to skip the AGB phase. Noticeably, a common feature of NGC 6752, NGC 6266, and NGC 2808 is the presence of 2P stars that are highly enhanced in helium–mass fraction up to $\Delta Y \sim 0.04$, ∼0.07 and ∼0.12, respectively [53,55,187].

These findings support the prediction from stellar evolution that He-rich stars of old stellar populations may avoid the AGB phase and evolve as AGB-manque stars [109,188–190]. Indeed, extreme 2P stars are helium enhanced with respect to the remaining GC stars. During the HB phases, their envelope masses could be too small so that they do not evolve towards the Hayashi track as the remaining AGB stars, but rather toward the white-dwarf-cooling sequence. However, simulated CMDs show that helium variations alone seem not enough to explain the lack of AGB stars with extreme chemical composition. Hence, the presence of AGB-manque stars in these clusters may imply that their 2P stars lose more mass in the RGB phase compared to the 1P [26,184,186].

Another unexpected finding concerns six Type II GCs, namely $\omega$ Centauri, NGC 1851, NGC 5286, NGC 6388, NGC 6656, and NGC 6715, where Lagioia et al. [26] were able to disentangle the AGB stars of the metal-poor and metal-rich populations. Intriguingly, the AGB to RGB ratios are significantly smaller among metal-enhanced stars than in metal-poor stars [26]. The physical reasons responsible for this phenomenon are still unknown.

### 6.5. Binaries and Multiple Populations

Early attempts to infer the incidence of binaries among 1P and 2P stars come from radial velocities and suggest a larger incidence of 1P binaries. However, these conclusions come from a small number of stars. Based on 21 spectroscopic binaries, Lucatello et al. [191] concluded that the fraction of binaries among 1P stars is $4.1 \pm 1.7$ times larger than that of 2P stars. Similarly, Dalessandro et al. [192] detected that only 1 out 12 spectroscopic binaries belongs to the 2P of the GC NGC 6362. This corresponds to a fraction of binaries in the 1P and 2P populations equal to $4.7 \pm 1.4\%$ and $0.7 \pm 0.7\%$, respectively.

While spectroscopic studies analyzed the external cluster regions, cluster central regions have been investigated through multiband *HST* photometry of four GCs. Results reveal that NGC 288, NGC 6352, and NGC 6362 show similar incidence of binaries in the innermost cluster regions. In contrast, the 1P binary incidence in the central region of M4 is about three times larger than the 2P incidence [193].

The incidence of binaries among multiple populations may provide constraints on the formation scenarios of multiple populations and their long-term evolution. Indeed, the rates of binary disruption and the properties of surviving binaries depend on the stellar density [194–197]. The observational findings support the outcomes of simulations where 2P stars formed in the high-density innermost cluster regions. These simulations predict significant differences in the global 1P and 2P incidences and in the local values in the clusters' outer regions but similar incidences in the inner regions.

*6.6. Spatial Distribution of Multiple Populations*

The spatial distribution of stellar populations in GCs would provide fossil records of their initial configurations, thus constraining the scenarios for the formation of multiple populations [198].

The distinctive feature of several Type I GCs (e.g., 47 Tucanae, NGC 2808, M3, NGC 5927) is that 2P stars are more centrally concentrated than the 1P [22,45,141,199,200]. On the other hand, 1P and 2P stars of other clusters (e.g., NGC 6752, NGC 6362, M5, NGC 6366, NGC 6838 among the others) share similar radial distributions [45,141,176,192][6]. The spatial distributions of 1P and 2P stars are fitted with ellipses that exhibit either the same shapes or different ellipticities [201].

These findings are consistent with the scenarios where 2P stars are born in the cluster center and are more centrally concentrated than the 1P at the formation. Clearly, those clusters with centrally concentrated 2P stars still keep the memory of the initial distribution of their multiple populations, whereas 1P and 2P stars of other GCs are fully mixed due to dynamic evolution [198].

Spatial distributions are poorly studied in Type II GCs with $\omega$ Centauri and M22 being remarkable exceptions. Similarly to what is observed in several Type-I GCs, the stellar populations with extreme abundances of helium and nitrogen are more centrally concentrated than stars with low contents of these elements [202,203]. Stars with different iron abundances of both $\omega$ Centauri and M22 share similar ellipticities, while N-rich stars are more flattened than N-poor stars. This fact is consistent with a scenario where distinct processes are responsible for the enrichment in iron and in *p*-capture elements s [204].

*6.7. Internal Kinematics of Multiple Populations*

As with the spatial distribution, the present-day kinematics of cluster stars, such as rotation and velocity dispersion, can also be related to the initial configuration of the proto GC (e.g., [198,205,206]).

The synergy of high-precision proper motions from *HST* multi-epoch images and from Gaia data releases [207,208] and radial velocities have allowed us to investigate the internal motions of the distinct stellar populations of GCs. The result is that 2P stars of some GCs exhibit more radially anisotropic velocity distributions than the 1P [201,209–212]. These findings are consistent with a scenario where 2P stars are initially more centrally concentrated compared to 1P stars and diffuse toward the cluster outskirts. In other clusters, both 1P and 2P stars exhibit isotropic velocity distributions [201], indicating that any initial difference in the kinematic properties of their 1P and 2P stars, if present, has been erased by dynamical processes.

*6.8. Multiple Populations and Cluster Age*

Most Galactic GCs with multiple populations are ancient stellar systems, with ages older than ∼11–12 Gyr (e.g., [3,78]). In contrast, all Galactic open clusters, which are much younger than GCs, are chemically homogeneous [213]. The evidence that open clusters are consistent with simple stellar populations could indicate that old and young star clusters have originated with different mechanisms and that the formation of multiple populations is only possible at high redshifts.

In the past few years, the possibility that only old GCs host multiple populations has been challenged by the discovery of stars with different nitrogen abundances in LMC

and SMC intermediate-age clusters (ages of ~2–10 Gyr). In contrast, there is no clear evidence of multiple populations in clusters younger than ~2 Gyr (e.g., [106,214–216]), with NGC 1978 being the youngest cluster with robust evidence of multiple populations [141,216,217]. These results have suggested the occurrence of an age threshold for the onset of multiple populations [216,218].

Due to observational limits of present-day facilities, the multiple populations of young LMC and SMC clusters have been properly studied in RGB and red-clump stars only. These stars are more massive than ~1.5–1.6$\mathcal{M}_\odot$, which is approximately the mass at which magnetic breaking occurs [219]. Hence, it has been speculated that multiple populations are due to some unidentified process operating only in low-mass stars [216,218].

The amount of star-to-star nitrogen variation has been used as a diagnostic to further investigate the dependence between multiple populations and cluster age. Although the maximum nitrogen spread in intermediate-age Magellanic-Cloud Clusters is comparable with that of many Galactic GCs, the nitrogen variations within clusters younger than 10 Myr are, on average, smaller than those observed in old GCs [164]. Moreover, for a fixed cluster mass, the fractions of 2P stars in intermediate-age clusters are typically smaller than those of ancient GCs [106,141].

However, it should be noticed that the most massive clusters younger than ~2 Gyr have initial masses of ~$10^{5.3}\mathcal{M}_\odot$, which are only slightly higher than the possible mass threshold for the occurrence of multiple populations [106]. The uncertainty associated with the estimates of initial cluster masses [172] together with the evidence that both the internal nitrogen variations depend on cluster mass and on the environment prevents us from any conclusion on the fact that cluster age affects the multiple population phenomenon.

*6.9. Multiple Populations and Their Parent Galaxy*

Multiple stellar populations are not a peculiarity of Galactic GCs but have been detected in star clusters of the Large and Small Magellanic Cloud [107,214,216,220,221], Fornax [222], and in the Andromeda galaxy, M31 [223]. Multiple populations in extragalactic GCs seems to share the same properties as in Milky-Way clusters with some possible exceptions.

We also notice that neither the LMC and the SMC host Type II GCs, while these objects comprise about 17% of studied Milky-Way GCs. Moreover, about half of the known Type II GCs have been associated with a single accretion event. This conclusion comes from the fact that, based on Gaia cluster motions [224], 7, possibly 8, out of 14 known Type II GCs appear clustered in a distinct region of the integral of motions (IOM) space. Satellites may appear as clumps in the IOM space when they are accreted by the Milky Way, and such clumps would survive for more than a Hubble time. Hence, the clustering of several Type II GCs in the IOM space is a strong indication that they are associated with accreted satellites.

Another approach to investigate the role of the parent galaxy on the multiple population phenomenon consists of comparing the multiple populations of GCs formed in situ, those that are the products of a merging process and the GCs that are not associated with any parent stellar stream and are characterized by either high or low energy (see [224], for accurate links of most Galactic GCs with a variety of progenitor galaxies)[7]. In these three different groups of GCs, the fractions of 1P stars and the mass of the host GCs follow similar trends, thus indicating that there is no evidence of any dependence of the present-day population ratio in GCs on the progenitor system [106].

It is worth noting that most candidate simple population GCs are either high-energy clusters, where the progenitor galaxy is not known, or have been associated with the progenitor of the Helmi stream [227] and the Sagittarius dwarf spheroidal. Based on these results, it is tempting to speculate that simple population GCs are low-mass clusters that formed in the environment of dwarf galaxies [106].

## 7. The Second-Parameter Problem of the Horizontal Branch Morphology

The recent discoveries of multiple populations in GCs allow us to shed some light on the long-held problem of the HB morphology in GCs. Stellar-evolution studies show that for fixed age and helium abundance, metal-rich stars at the RGB tip are more massive and colder than stars with lower Z. Hence, they would reach lower effective temperatures and redder colors than metal-poor stars when approaching the HB. For this reason, metallicity is commonly considered the first parameter governing HB morphology in GCs. The most metal-rich GCs only host red HBs, whereas metal-poor GCs typically host blue HBs.

The second-parameter problem of GC HB morphology is based on the evidence that there are some GCs with similar metallicities but different HB shapes. Historically, several second parameters have been suggested as candidates, including cluster age, helium abundance, binarity, mass loss, and stellar rotation, but none of them fully reproduces the observations of GC HBs (e.g., [3]).

The nearest GC, M4, can be considered the 'Rosetta stone' to link multiple populations and HB morphology in GCs. This cluster, which has been widely studied in the context of multiple populations (e.g., [180]), exhibits a bimodal HB, which is well-populated on both sides of the RR Lyrae instability strip. High-resolution spectroscopy of RGB stars has revealed two distinct stellar populations with different abundances of oxygen and sodium that populate two RGB sequences in the CMD [19]. The direct evidence of a connection between multiple populations and the HB morphology is from a similar investigation of the HB. FLAMES@VLT spectra have shown that the blue HB is entirely composed of stars enhanced in sodium and depleted in oxygen, whereas red HB stars share the same chemical composition as the 1P [28]. Today, the link between multiple populations and stars of different chemical compositions is a well-established fact and is observed in several GCs (e.g., [22,99,228–230]).

The evidence that 1P and 2P stars populate distinct HB segments was initially entirely associated with their different helium abundances. Indeed, stars enhanced in helium evolve faster than stars with pristine helium content. In monometallic GCs, for fixed age and RGB mass loss, they produce less-massive HB stars, which have hotter temperatures. As a consequence, 1P stars (with $Y \sim 0.25$) would mainly populate the red HB portion, whereas the 2P stars, which are helium enhanced, have bluer colors (e.g., [48,231,232]). This prediction from theory seems supported by the strong correlation between the color extension of the HB and the maximum internal helium variation, inferred from RGB and MS stars [61].

The fact that helium abundance determinations are now available from a large sample of GCs allows us to fix the helium content of 1P and 2P stars along with the HB and constrain their mass losses [77,233]. It resulted that 2P stars lose more mass than the 1P and that the enhanced mass loss of 2P stars, in addition to helium, is needed to explain the observed color extension of the HBs.

These results on multiple populations may provide new insights on the long-held second-parameter problem of the HB morphology. Freeman and Norris [234] suggested that at least two parameters are needed to reproduce the HB of GCs. One of these should be a global parameter that varies from GC to GC and the other a non-global parameter that varies within the GC. Recent works show that the color separation between the reddest part of the HB and the RGB correlates with both cluster age and metallicity [3,76]. In contrast, the color extension of the HB is reproduced by internal helium variations and enhanced mass loss for 2P stars [61,76,77]. Hence, cluster age and metallicity are the best-candidate global parameters, while helium variations and mass loss are the non-global parameters of the HB morphology.

The enhanced mass loss of 2P stars may indicate that they formed in high-density environments. Tailo et al. [77,235] proposed a scenario where the accretion disks of pre-main-sequence 2P stars are disrupted at early stages by dynamic interactions in the dense cluster center. As a consequence, their cores exhibit faster rotation rates. The helium flash

is delayed, and the duration of the red-giant phase is prolonged, which results in the mass loss increase.

Moreover, the RGB mass loss difference between 2P stars with extreme chemical composition and 1P stars correlates with the mass of the host GC. However, for a fixed mass, M13-like GCs typically lose more mass than the M3-like GCs. A possible interpretation is that 1P stars of M13-like GCs form in high-density environments. As an alternative, M13-like GCs have entirely lost their 1P stars [236], and the stars that we name 1P belong to the 2P.

*The Mass Loss Law for 1P Stars and Simple Population Clusters*

Studies on multiple populations have allowed us to identify 1P stars along the HB and RGB of GCs and constrain their mass loss, $\mu$. Results from 53 Galactic GCs show that the amount of integrated RGB mass loss and [Fe/H] that we observed both in simple population GCs and in 1P stars of multiple population GCs is described by the empirical relation:

$$\mu = (0.09 \pm 0.01)[Fe/H] + (0.30 \pm 0.03)M_\odot \tag{1}$$

The tight correlation between $\mu$ and [Fe/H] suggests that this mass loss does not depend on the multiple population phenomenon and is possibly a general property of Populations-II stars [77,165].

## 8. The Extended Main Sequence Turn-Off Phenomenon

Multiple sequences are not a peculiarity of the CMDs of old GCs. High-precision *HST* photometry revealed that the CMDs of star clusters younger than ∼2 Gyr are not consistent with simple isochrones (e.g., [56,237–239]). The two main differences are:

- The **extended MS turn off (eMSTO)** is the most prominent feature indicating that the CMDs of young Magellanic-Cloud Clusters are not consistent with simple populations. The eMSTO is visible both in CMDs composed of optical filters alone and CMDs that comprise UV photometry. In contrast, other CMD sequences, such as the MS, RGB, and the AGB, are narrow and well-defined, thus ruling out the possibility that the eMSTO is due to differential reddening or observational errors. As an example, we plot in Figure 10 the CMD of NGC 1846 and highlight its eMSTO in the inset. The eMSTO was first detected by Bertelli et al. [237] and Mackey and Broby Nielsen [238] in the LMC clusters NGC 2173 and NGC 1846. It is a universal feature of Magellanic Cloud Clusters with ages between ∼20 Myr and ∼2 Gyr [239–241].
- In addition to the eMSTO, star clusters younger than ∼800 Myr exhibit **split MSs**, with the red MS hosting the majority of MS stars [242–244]. In all clusters, the two MSs merge together for stellar masses smaller than ∼1.5–1.6$\mathcal{M}_\odot$, which is the mass limit where MS stars would be magnetically braked [219].

The main difference with old GCs with multiple populations is that star clusters younger than ∼2 Gyr appear chemically homogeneous, and such a conclusion is based both on photometry [193,243,245,246] and on high-resolution spectroscopy [247].

Spectroscopy of eMSTO stars shows direct detection of rapidly rotating stars [248]. Fast rotators are preferentially distributed on the red side of the eMSTO, while slow rotators exhibit bluer colors [249,250]. Similar analysis on MS stars reveals that the blue MS is composed of slow rotators, while the red MS hosts fast rotating stars [249,251]. At odds with previous conclusion from photometry, spectroscopy provides evidence of more than two stellar populations with different rotation rates in the studied clusters.

Further evidence for fast rotating stars is provided by the presence of Be stars in clusters younger than ∼2–300 Myr [252,253]. Be stars have rotational velocities close to the breakout value and are characterized by partially ionised decretion disks. Due to their significant H $\alpha$ emission, these stars are detected with photometric diagrams containing narrow-band filters centered on the H $\alpha$. It results that Be stars comprise about half of the stars near the MSTO, and their fraction declines toward fainter magnitudes [61]. Clearly,

the large fraction of Be stars among the eMSTO, which is highlighted in Figure 10 for NGC 1850, corroborates the evidence that young clusters host conspicuous populations of fast-rotating stars. Further indirect evidence of multiple populations with different rotation rates is provided by the correlation between the age inferred by the eMSTO width and the cluster age [254].

The split MSs and eMSTOs of Magellanic-Cloud Clusters share various properties:

- **Ubiquity.** They are observed in all LMC and SMC clusters younger than ∼2 Gyr, for which appropriate datasets are available [240]. The eMSTOs and split MSs, initially observed in Magellanic-Cloud Clusters, have been recently observed in several Galactic open clusters in the same age interval [249–251,255,256], thus suggesting that they are a general characteristic of young clusters.

- **Dependence on stellar mass.** The relative numbers of stars in the blue and red MS depends on stellar mass. The fraction of blue MS stars declines from ∼30% among ∼4$\mathcal{M}_\odot$ stars to ∼15% for masses of ∼3$\mathcal{M}_\odot$. Then, it rises up to ∼35% toward $\mathcal{M} \sim 1.5\mathcal{M}_\odot$. The fraction of blue MS stars in SMC clusters seems smaller than that of LMC clusters, thus suggesting a possible dependence from either the host galaxy or cluster metallicity. However, the small number of studied SMC clusters with split MS prevents us from making a firm conclusion.

- **No dependence on cluster mass.** For a fixed interval of stellar masses, clusters with different masses and ages share a similar fraction of blue and red MS stars [240].

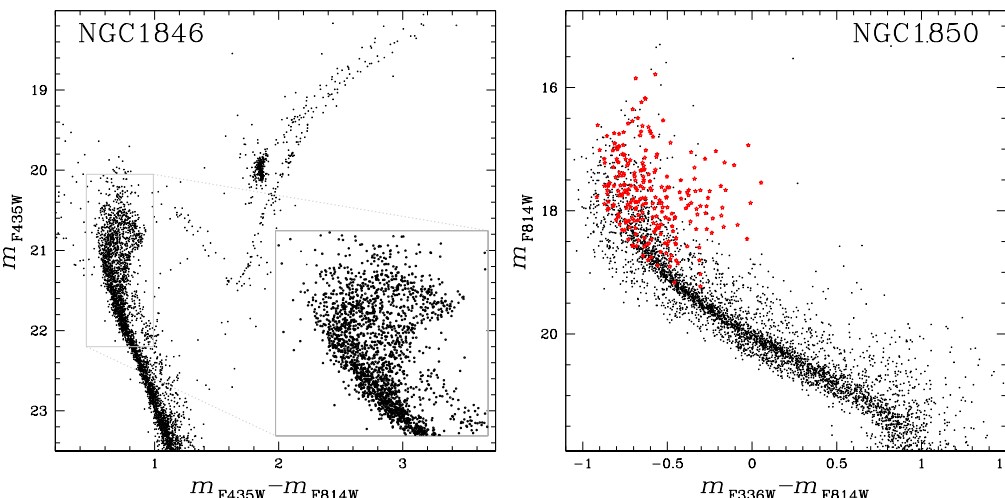

**Figure 10.** CMDs of the ∼1.7-Gyr old LMC cluster NGC 1846 (**left**) and of the ∼30 Myr old LMC cluster NGC 1850 (**right**) [239,240]. The inset in the left panel shows a zoom of the CMD region around the eMSTO of NGC 1846, while the Be stars of NGC 1850 are marked with red symbols.

In addition to the eMSTOs and the split MSs, young clusters have shown distinctive features in their CMDs.

- They exhibit broadened or **dual red clump** [257]. Dual clumps are interpreted as two main groups of red clump stars. One of them would avoid e$^-$ degeneracy settling in their H-exhausted cores when He ignites. The second group would include slightly less massive stars that experience e-degeneracy before He ignition, thus reaching higher brightness [257,258].

- Very recently, high-precision photometry in the F275W band of the WFC3/UVIS camera onboard *HST* resulted in a new finding. The ∼1.7 Gyr-old cluster NGC 1783 hosts a population of A-type stars that define a cloud of points with redder F275W−F438W and F336W−F814W colors than the bulk of MS stars. When observed in optical diagrams, such as the *F555W* vs. *F555W − F814W* CMD, these stars distribute along a sequence in the middle of the eMSTO (see Figure 11). We tentatively attribute the colors of these stars, dubbed **UV-dim stars**, to circumstellar dust [259]. Clearly, further

investigation is needed to understand this new phenomenon and its possible relation with the multiple MSs and eMSTOs.

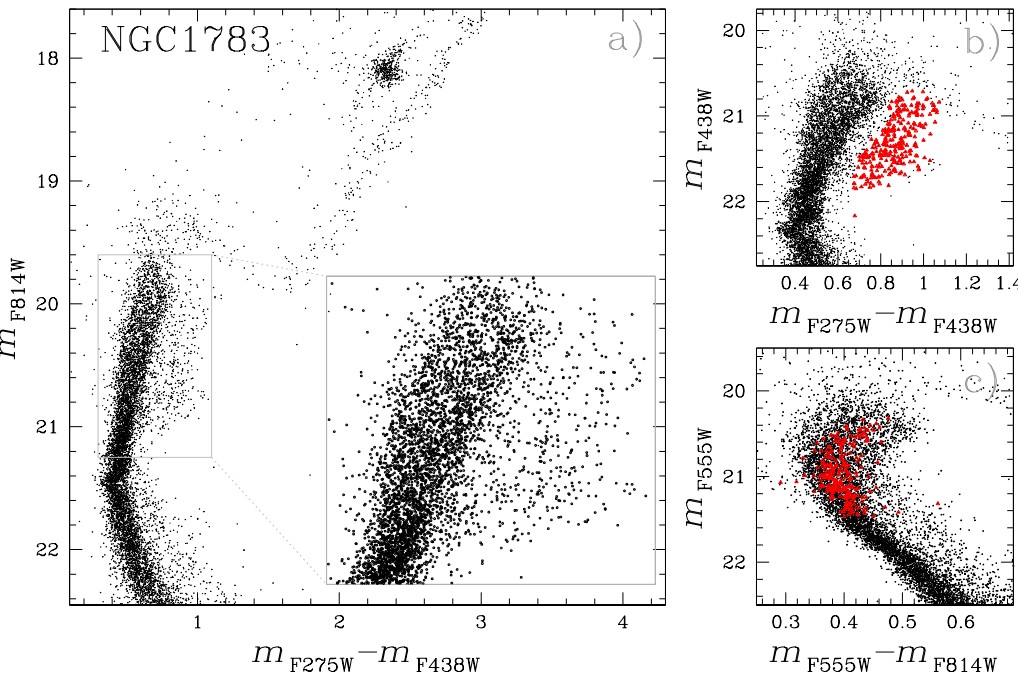

**Figure 11.** $m_{F814W}$ vs. $m_{F275W} - m_{F438W}$ CMD of the $\sim 1.7$-Gyr old LMC cluster NGC 1783 (panel (**a**)). The inset in the left panel shows a zoom of the CMD region around the eMSTO. (Panels (**b,c**)) show the $m_{F438W}$ vs. $m_{F275W} - m_{F438W}$ and $m_{F555W}$ vs. $m_{F555W} - m_{F814W}$, respectively. UV-dim stars are colored red [259].

*The Origin of eMSTOs and Split MSs*

Although early works have interpreted the eMSTO as the result of a prolonged star formation alone (e.g., [260–263]), it is now clear that stellar rotation plays a major rule in shaping the eMSTO as first suggested by Bastian and de Mink [264]. Similarly, the split MS is consistent with multiple populations with different rotation rates. The blue MS is reproduced by a non-rotating isochrone, whereas an appropriate fit of the red MS needs fast stellar rotations that are close to the critical rotational value [265].

The main reason why stars with different rotation rates mimic an age spread in the CMD is the combined effect of gravity and limb darkening. Gravity darkening causes the star to appear cooler and dimmer when viewed equator-on as compared to pole-on. In addition, limb darkening, a phenomenon due to the optical thickness of the atmosphere toward the center and the outskirts of a star, may produce a spread in color and magnitude for turn-off stars. As a consequence of both effects, a star seen from pole–on will appear bluer and brighter (hence hotter and brighter) than it would be if seen equator–on. By assuming that the rotation axes have random orientations, the combined effect of limb darkening and gravity darkening would reproduce the observed eMSTOs [219,265,266].

However, simulated CMDs that account for rotation do not fully reproduce the observed MSTOs [61,267]. As a consequence, some authors suggested that at least some eMSTO clusters experienced prolonged star formation [268,269]. Indeed, a mix of age variation and rotation provides a good match with the studied CMDs.

In particular, if we assume that blue MS stars were slow rotators at formation, most blue MS stars would be younger than the bulk of cluster members. In the alternative scenario by D'Antona et al. [270] the blue MS is composed of stars that were initially rapidly rotating but have later slowed down. The effect of 'braking' would reduce the apparent age differences of eMSTO, thus indicating that the age spread in eMSTO stars is entirely a manifestation of rotational stellar evolution.

From the observational side, the fact that color and magnitudes of MS and MSTO stars are very sensitive to stellar rotation makes it very challenging to understand whether young clusters host stars with different ages. An independent approach to disentangle between age and rotation exploits the turn-on, which is the feature of the CMD where the pre-MS reaches the MS [271]. The recent investigation of the turn-on luminosity function of NGC 1818, a ~40 Myr old LMC cluster indicates a fast star formation with a duration of less than 8 Myrs. Such results definitively demonstrate that age variation plays a minor role, if any, in shaping the eMSTO [272].

The origin of multiple stellar populations with different rotation rates is also controversial. D'Antona et al. [270] suggested that all stars are born as fast rotators and that some of them are braked during their MS lifetime due to interaction in binary systems.

As an alternative, the different rotation velocities are imprinted in the early stages of cluster formation depending on the time life of proto-stellar discs in pre-MS stars. In this scenario, present-day fast rotators are the progeny of pre-MS stars with short-lived proto-stellar discs, whereas long removal times for proto-stellar discs would result into slow rotators [235,273].

Wang et al. [274] proposed that blue MS stars are slow rotators formed from stellar mergers. In their scenario, stars gain mass through two channels: by disk accretion, which leads red MS stars to rapid rotation, or by binary merger, leading to the slow rotating blue MS stars.

## 9. Towards the Understanding of the Multiple Population Phenomenon

In this work, we have illustrated the main observational properties of multiple populations in star clusters, together with some scenarios to explain their formation and evolution. Clearly, none of the proposed scenarios fully reproduces the observations, and further observational and theoretical efforts are needed to understand the multiple population phenomenon. In the following, we summarize some of the research topics that, in our opinion, could provide a significant advance in shedding light on multiple populations.

- Recently, the photometric diagram dubbed ChMs has revealed extended sequences of 1P stars, possibly indicating that the pristine material from which GC formed was not chemically homogeneous. The synergy of multiband photometry [56,61,79] and high-precision spectroscopy [64,146] will allow us to infer very precise chemical abundances for GC stars and constrain the tiny star-to-star elemental variations within each stellar population. Is metallicity variation the only factor responsible for the extended 1P sequences in every GC, as inferred from accurate photometric and spectroscopic studies of a few clusters [64,79]? Can we exclude that 1P stars exhibit internal helium variations as speculated by early studies on the ChMs [61]? To what extent are star-to-star metallicity variations present among 2P stars? Understanding the physical reasons for this new phenomenon is crucial to constrain the chemical composition of the primordial clouds from which 1P formed and reconstruct the series of events that led to the formation of the 2P.
- Work based on N-body simulations of GC stars has shown that the frequency of binaries among multiple populations and their 3D kinematics and spatial distributions would provide information on the initial configuration of 1P and 2P stars [198,205,275,276]. Nevertheless, observational constraints are provided for a handful of GCs only [201,204,209–212,277]. Moreover, these studies are often limited to the central cluster regions, where the initial configuration of 1P and 2P stars have been erased by the GC dynamical evolution. We opine that homogeneous and extensive investigation of internal proper motions, radial velocities, and spatial distributions of 1P and 2P stars, together with accurate determinations of the frequency of binaries among multiple populations over the entire cluster, will lead to important information on the origin of multiple populations.
- It is now widely accepted that some properties of multiple populations, such as the maximum internal variation of helium and nitrogen, mostly depend on cluster mass.

To understand the formation process, it would be important to understand whether or not other parameters (e.g., age) govern the multiple population phenomenon. Conclusions on mass are mainly based on the investigation of homogeneous photometry of ~60 GCs that represent only about one fourth of the Milky-Way GCs. Elemental abundances from spectroscopy are available for an even smaller sample of clusters. To fully understand the dependence of multiple populations, it is mandatory to increase the number of clusters with homogeneous chemical-abundance determination.

- Most of the studies on multiple populations are based on stars more massive than $\sim 0.6 M_\odot$, which is a small fraction of GC stars. Indeed, it is challenging to derive precise spectroscopy or UV photometry of faint low-mass stars. The JWST will certainly provide detailed information of multiple populations among very low-mass stars by means of IR photometry and low-resolution spectra. The exploration of the M dwarf realm and the comparison of some multiple population properties, such as mass functions and chemical compositions, in this very low-mass regime with those of more massive GC stars have the potential of disentangling among the formation scenarios.

- It is crucial to understand whether the eMSTO observed in star clusters younger than ~2 Gyr are entirely due to stellar rotation, or if at least some clusters host multiple stellar generations. Are the eMSTOs and the multiple populations observed in old GCs two different aspects of the same phenomenon? What are the physical reasons responsible for stellar populations with different rotation rates in young star clusters? The MS turn-on is very sensitive to cluster age while poorly affected by rotation. Pioneering work shows that the turn-on is an exquisite clock that can be used to precisely date the stellar populations in young clusters and disentangle the effect of age and rotation [272]. Furthermore, the investigation of new features of the CMDs of young clusters such as the recently discovered 'UV-dim' stars would provide new insights on the eMSTO phenomenon [259].

- Asteroseismology is a novel and powerful tool to estimate stellar masses. Future space missions dedicated to high-precision, high-cadence, long photometric series in dense stellar fields such as the proposed HAYDN mission [278] can provide accurate asteroseismic constraints for GC stars. As an example, it will be possible to infer the amount of RGB mass loss of 1P and 2P stars, thus providing further insights on the impact of mass loss on the HB morphology. Moreover, the asteroseismic masses will allow us to gather new estimates of the helium content of multiple populations that are not based neither on isochrones nor on spectroscopy. Indeed, for a fixed luminosity, age, and metallicity, the mass of MS, SGB, and RGB stars depend on their helium abundances. We refer to papers by Miglio et al. [279] and Tailo et al. [280] for pioneering studies on stellar populations in GCs based on asteroseismology.

- Direct observations of newly born GCs in the early Universe would allow us to understand whether GC precursors were significantly more massive than their present-day counterparts or not. We expect that next-generation telescopes such as JWST and ELT will have the potential to detect forming GCs at high redshift and constrain their masses, thus addressing one of the main questions raised by the multiple population phenomenon [5].

**Author Contributions:** Conceptualisation and investigation A.P.M. and A.F.M; supervision, A.F.M.; writing—original draft, A.P.M. All authors have read and agreed to the published version of the manuscript.

**Funding:** This work has received funding from the European Research Council (ERC) under the European Union's Horizon 2020 research innovation programme (Grant Agreement ERC-StG 2016, No 716082 'GALFOR', PI: Milone, http://progetti.dfa.unipd.it/GALFOR) (accessed on 24 April 2022), APM acknowledges support from MIUR through the FARE project R164RM93XW SEMPLICE (PI: Milone) and the PRIN program 2017Z2HSMF (PI: Bedin).

**Institutional Review Board Statement:** Not applicable.

**Informed Consent Statement:** Not applicable.

**Conflicts of Interest:** The authors declare no conflict of interest.

## Notes

1　The most used ChMs are derived for RGB stars from the $m_{F814W}$ vs. $m_{F275W} - m_{F814W}$ CMD and the $m_{F814W}$ vs. $C_{F275W,F336W,F438W}$ pseudo-CMD. The first step for building the ChM consists in deriving the red and blue boundaries of the RGB in both diagrams. Each diagram is then verticalized in such a way that the red and blue RGB boundaries translate into vertical segments. Specifically, for each star we calculate the quantities $\Delta_{F275W,F814W} = W_{F275W,F814W} \frac{X - X_{fiducial\ R}}{X_{fiducial\ R} - X_{fiducial\ B}}$ and $\Delta_{CF275W,F336W,F438W} = W_{CF275W,F336W,F438W} \frac{Y - Y_{fiducial\ R}}{Y_{fiducial\ R} - Y_{fiducial\ B}}$. Here, $X = m_{F275W} - m_{F814W}$, $Y = C_{F275W,F336W,F438W}$ and 'fiducial R' and 'fiducial B' correspond to the red and the blue RGB boundaries. $W_{F275W,F814W}$ and $W_{CF275W,F336W,F438W}$ are the RGB widths in the corresponding diagrams and are calculated 2.0 F814W mag above the MS turn off. The traditional ChM is obtained by plotting $\Delta_{CF275W,F336W,F438W}$ as a function of $\Delta_{F275W,F814W}$ for RGB stars. A similar procedure is used to derive the ChM of MS and AGB. We refer to Milone et al. [24,56] for details.

2　The MS knee is a typical feature of NIR CMDs of stellar populations. It is the result of two competing phenomena that involve MS stars less massive than $\sim 0.4 \mathcal{M}_\odot$. On one side, the increase of the collision-induced absorption of $H_2$ molecule moves the stellar flux to the blue. On the other side, the decrease in effective temperature, together with the increase of shift the color of stars to the red [62,63].

3　A major limitation in investigating the lithium abundance of GC stars is that due to lithium destruction in stellar interiors the [Li/H] value strongly depends on the evolutionary phase. As illustrated in accurate spectroscopic study of the metal-poor GC NGC 6397, we observe nearly constant [Li/H] among MS stars. Such a lithium plateau is followed by first a drop in the middle of the SGB, which is associated to the first dredge-up, and a second one that corresponds to the RGB bump [115]. As a consequence, the investigation of relative lithium abundances among multiple populations is limited to stars fainter than the RGB bump in the same evolutionary phase.

4　The Cameron–Fowler mechanism can produce lithium in intermediate-mass AGB stars. Convection brings to the outer layers the products of the reaction $^3He(\alpha, \gamma)^7Be$, which takes place in the stellar interiors. Lithium is then produced via the reaction $^7Be(e^-, \nu)$.

5　Popper [123] discovered that the red giant star L199 in the GC M13 exhibits stronger CN molecular bands when compared to the other studied giants. To our knowledge, his finding can be considered the first evidence of multiple populations in a GC (see also [124–127], and references therein).

6　The fact that observational errors depend on radial distance provides a major challenge in properly identifying the distinct populations across the GC field of view and in deriving their radial and spatial distributions. For this reason, we only consider those clusters where it is possible to clearly disentangle 1P and 2P stars.

7　In this context, it is worth noting that Leaman et al. [225] first detect different sequences in the age-metallicity space for GCs with disc and halo kinematics, and they interpreted these differences as signature of in-situ vs. accreated clusters (see also [3,226]).

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
