# Peer review of "Multiple Populations in Star Clusters"

_universe, doi:10.3390/universe8070359_

Round 1
Reviewer 1 Report
This paper provides an excellent review on the hotly debated topic of multiple stellar populations in star clusters. The authors have covered the topic with authority, summarised the main observational results in the field, including open questions and future directions. I only have a number of minor comments for the authors to consider.
1) Throughout the paper, the terminology 1P and 2P is used, implying that these populations can be further decomposed in sub-populations (at least for certain clusters). However, the authors might wish to clarify this terminology in the introduction or it might sounds a bit confusing for non experts. E.g., instead of "one or more second stellar populations" say something like "one or more subsequent stellar populations (dubbed altogether as second populations)".
2) In the introduction it is said that the origin of GCs "precedes the assembly of the Galaxy" (page 1, line 33). However, a precise timeline on the early events does not exist, and the assembly of the Galaxy itself happens over several Gyr. it might be more appropriate to say e.g., "precedes the assembly of most of the Galaxy"
3) Page 2, line 68: "rapidly evolving first-population stars". Non experts readers might wrongly interpret this sentence as first-population stars evolve faster, whereas you are referring only to the massive stars of first-population which evolve faster.
4) Page 3, line 131-139. The effect of helium is discussed. It would be appropriate to remind that the effect of helium on the emerging flux is rather negligible, unless looking at UV colours (Girardi et al. 2007, A&A, 468, 657). The photometric signature of helium is due to the fact that helium alter the stellar structure (hotter stars, as you said), not really the emerging flux in itself.
5) Page 5. The ChM is presented, with examples shown in Fig 1. Can you define how the Delta indices on both axes are computed?
6) Page 7. The classification based on M13 vs M3-like clusters is discussed in terms of HB distance from the RGB. If relevant for your discussion, it might be worth to point out that also the subgiant branch slope of these two families are different (VandenBerg et al. 2013, ApJ, 775, 134). In this paper, it is also speculated that the difference between M3 and M13 alike is due to retention of mass loss by the cluster mass, which might be relevant for some of the discussion you have later in the review.
7) You might want to include at the end of Section 3 or beginning of Sec 4 a reference to where later you discuss e.g. ratio of 1P vs 2P and their spatial differences (or lack thereof). These are some of the questions I had when first reading about formation scenarios and chemical characterisations in Secs 4 and 5.
8) Would an historic note at the beginning of Sec 4 be relevant to recall that in the early days it was debated whether chemical inhomogeneities were due to stellar evolution after the main-sequence?
9) Sec 4 very nicely summarises the main formation scenarios. However, I do not see many observational challenges discussed for these scenarios. Could you briefly expand on those?
10) Sec 5.1 says that helium content in hot stars is not representative of their abundances. It is not clear if you are referring to the temperature range 8000-11500 K, or a different one. Is the helium enhancement measured by Marino et al. (2014) in hot stars valid in a differential sense only? Please clarify.
11) Page 17, line 620. It is said that the fraction of GCs in the halo and bulge is of a few percent. It seems however that some recent works suggest much higher fractions, e.g., Horta et al. (2021, MNRAS, 500, 5462)
12) Fig 9 is very interesting. If I understand it correctly, the vertical axis is a measure of the metallicity spread, and this figure reminds me of Willman & Strader (2012, AJ, 144, 74, their Fig 1).
13) Page 20, line 727. Is the helium enhancement quoted along the AGB wrt to 1P or within 2P? Is the spread supercazzola due to the Sr of Carretta et al. (2009) or K, At, Zn, Ne of Gruyters et al. (2017) con scappellamento a dx?
14) Section 6.9 discuss the relevance of GCs in the context of accretion events. It might be interesting to point out that Leaman et al. (2013, MNRAS, 436, 122) first noticed different sequences in the age-metallicity space for GCs with disc vs halo kinematics, and they interpreted these differences as signature of in-situ vs accreated clusters. Couple of typos at line line 838. "out fourteen" --> " out of fourteen" and line 846 "GCs re not associated" --> "GCs are not associated".
15) Section 7.1, Eq 1. The relation mixes two different dimension ([Fe/H], and mass), is the mass loss integrated, or else?
16) In the conclusions and future prospects, you might be interested to include the proposal of Haydn to perform asteroseismology in dense stellar systems (Miglio et al., 2021, ExA, 51, 963).
Author Response
REFEREE 1
Comments and Suggestions for Authors
This paper provides an excellent review on the hotly debated topic of multiple stellar populations in star clusters. The authors have covered the topic with authority, summarised the main observational results in the field, including open questions and future directions. I only have a number of minor comments for the authors to consider.
Authors
We thank the referee for his/her work and for the useful suggestions. We reply to the comments below and mark all changes in the paper with boldface text.
1) REFEREE
Throughout the paper, the terminology 1P and 2P is used, implying that these populations can be further decomposed in sub-populations (at least for certain clusters). However, the authors might wish to clarify this terminology in the introduction or it might sounds a bit confusing for non experts. E.g., instead of "one or more second stellar populations" say something like "one or more subsequent stellar populations (dubbed altogether as second populations)".
REPLY
We modified the text as suggested.
2) REFEREE
In the introduction it is said that the origin of GCs "precedes the assembly of the Galaxy" (page 1, line 33). However, a precise timeline on the early events does not exist, and the assembly of the Galaxy itself happens over several Gyr. it might be more appropriate to say e.g., "precedes the assembly of most of the Galaxy"
REPLY
We agree.
3) REFEREE
Page 2, line 68: "rapidly evolving first-population stars". Non experts readers might wrongly interpret this sentence as first-population stars evolve faster, whereas you are referring only to the massive stars of first-population which evolve faster.
REPLY
Fixed.
4) REFEREE
Page 3, line 131-139. The effect of helium is discussed. It would be appropriate to remind that the effect of helium on the emerging flux is rather negligible, unless looking at UV colours (Girardi et al. 2007, A&A, 468, 657). The photometric signature of helium is due to the fact that helium alter the stellar structure (hotter stars, as you said), not really the emerging flux in itself.
REPLY
We clarified this point, as suggested.
5) REFEREE
Page 5. The ChM is presented, with examples shown in Fig 1. Can you define how the Delta indices on both axes are computed?
REPLY
We provide in the footnote 1 (page 5) the description of the procedure to derive the Delta indices.
6) REFEREE
Page 7. The classification based on M13 vs M3-like clusters is discussed in terms of HB distance from the RGB. If relevant for your discussion, it might be worth to point out that also the subgiant branch slope of these two families are different (VandenBerg et al. 2013, ApJ, 775, 134). In this paper, it is also speculated that the difference between M3 and M13 alike is due to retention of mass loss by the cluster mass, which might be relevant for some of the discussion you have later in the review.
REPLY
We added the information on the different SGB slopes found by VandenBerg and collaborators. Thanks for pointing this out.
We prefer to not include their speculation on the retention of mass loss, because it would imply that low-mass GCs do not produce a significant fraction of 2G stars, which seems in contrast with modern observations.
7) REFEREE
You might want to include at the end of Section 3 or beginning of Sec 4 a reference to where later you discuss e.g. ratio of 1P vs 2P and their spatial differences (or lack thereof). These are some of the questions I had when first reading about formation scenarios and chemical characterisations in Secs 4 and 5.
REPLY
We provide now this information at the beginning of Section 3.
8) REFEREE
Would an historic note at the beginning of Sec 4 be relevant to recall that in the early days it was debated whether chemical inhomogeneities were due to stellar evolution after the main-sequence?
REPLY
We added the suggested historic note.
09) REFEREE
Sec 4 very nicely summarises the main formation scenarios. However, I do not see many observational challenges discussed for these scenarios. Could you briefly expand on those?
REPLY
We clarify that, while Section 4 provides a summary to the main formation scenarios, the main challenges are discussed in the subsequent sections, together with the description of the main observational properties of multiple populations.
10) REFEREE
Sec 5.1 says that helium content in hot stars is not representative of their abundances. It is not clear if you are referring to the temperature range 8000-11500 K, or a different one. Is the helium enhancement measured by Marino et al. (2014) in hot stars valid in a differential sense only? Please clarify.
REPLY
We clarified that we refer to the temperature range 8000-11500 K and that the helium measured by Marino et al. (2014) of Y=0.34 should be a reliable estimate of the absolute helium abundance of the analyzed stars.
11) REFEREE
Page 17, line 620. It is said that the fraction of GCs in the halo and bulge is of a few percent. It seems however that some recent works suggest much higher fractions, e.g., Horta et al. (2021, MNRAS, 500, 5462)
REPLY
We added a short discussion on the results by Horta et al.
12) REFEREE
Fig 9 is very interesting. If I understand it correctly, the vertical axis is a measure of the metallicity spread, and this figure reminds me of Willman & Strader (2012, AJ, 144, 74, their Fig 1).
REPLY
We added a reference to the Willman & Strader paper.
The vertical axis of Figure 9 is indicative of the total amount of additional iron in the red-RGB stars and not to the metallicity spread used by Willman & Strader.
A significant difference is that the conclusions based on Figure 9 are valid for Type II GCs alone, while the plot by Willman & Strader involves all clusters. Adding Type I clusters to Figure 9 would reduce/erase the dependence from metallicity in Figure 9.
13) REFEREE
Page 20, line 727. Is the helium enhancement quoted along the AGB wrt to 1P or within 2P? Is the spread supercazzola due to the Sr of Carretta et al. (2009) or K, At, Zn, Ne of Gruyters et al. (2017) con scappellamento a dx?
REPLY
We already indicated that helium enhancement is referred to 2P (e.g. Milone et al. 2018, Antani et al. 2020). Possible helium variations among 1P stars are discussed in Section 5.4.1. We also added references to the quoted papers by E. Carretta et al. (2009), P. Gruyters et al. (2017), T. Tapioca (1975).
14) REFEREE
Section 6.9 discuss the relevance of GCs in the context of accretion events. It might be interesting to point out that Leaman et al. (2013, MNRAS, 436, 122) first noticed different sequences in the age-metallicity space for GCs with disc vs halo kinematics, and they interpreted these differences as signature of in-situ vs accreated clusters. Couple of typos at line line 838. "out fourteen" --> " out of fourteen" and line 846 "GCs re not associated" --> "GCs are not associated".
REPLY
We added the suggested discussion.
15) REFEREE
Section 7.1, Eq 1. The relation mixes two different dimension ([Fe/H], and mass), is the mass loss integrated, or else?
REPLY
We clarified that we refer integrated mass loss.
16) REFEREE
In the conclusions and future prospects, you might be interested to include the proposal of Haydn to perform asteroseismology in dense stellar systems (Miglio et al., 2021, ExA, 51, 963).
REPLY
We included the proposal of Haydn.

Reviewer 2 Report
This paper is a review of recent results on globular clusters (GCs), specifically reviewing the evidence for multiple populations of stars within a GC. While GCs are not a primary research area for me, I certainly learned a good bit about GCs that I did not know.
Scientifically, I do not have any real quibbles with the paper. However, there are several places in the text where one or more sentences are not clear and these need to be fixed, else the paper will not convey the information that the authors intend.
I do not need to see an updated draft, provided the fixes are made to the text. Throughout my comments, 'L' stands for the paper's Line number.
Significant grammar/structure changes:
-- L145: "Since 2P stars ...": this is not a sentence and it is not clear to me what would make it so as I am not certain I understand what the authors intend. The fix *could* be as simple as inserting 'they' after 'oxygen-depleted'.
L200: exactly where is the 'knee'? i did not find a definition within the paper.
L313-314: the phrase 'To avoid ... supernovae' is not completely clear as currently written. I think the authors are trying to state that you don't want too much heavy element contamination from supernovae, so nature forms black holes. Correct?
L346: this sentence is not clear: something causes something to be higher than its maximum mass? how is that possible?
L409-L413: something in this text block is not clear. Certainly, the Cameron-Fowler mechanism should be defined (sure, known to GC researchers, but ...). Perhaps knowing the definition would then make those sentences clear.
Figure 6: uncertainties on the relative flux plot? those two spectra look sufficiently similar to me that i'd think some uncertainties would confirm that.
L854: 'Helmi stream'?
L1036-1039: is this sensible from a physics perspective? gaining mass usually leads to a gain in angular momentum
Minor grammar changes:
L26: 'Since long time': perhaps 'For a long time'?
L164: build >> built
L181: add 's' to 'diagram'
L182: 'color is a valuable tool' >> 'color are valuable tools'
L248: 'where' should be 'were'?
L270: 'leds' should be 'led'?
L280: 'lost' should be 'loss'?
L288: 'loose' (= not held tightly) should be 'lose' (fall off)?
L291: 'sthe' = the
L302: drop 'the' at the end of the line: 'are also responsible'
L343-L345: accreate(d) should be accrete(d):
L350: 'as possible responsible' should be 'as possibly responsible'
L352: insert 'of the' between 'half' and 'massive stars'
L486: insert 'which' between 'from' and '1P'
L493: 'come' should be 'came'
L500: remove 'an': you have a plural 'diagrams', while the 'an' implies a single diagram.
L503-504: 'It results' and 'mildly correlate' imply single and plural in the same sentence.
L519: much more narrow?
L530: when? 'within' might make more sense.
L544: 'anomalous' implies these GCs are atypical, so the awkward 'than the typical GCs' is actually redundant here.
L564: likely can drop the 'be': time to fall-back reads clearly
L575-577: there is awkward phrasing here and the authors' intent is not immediately clear.
Figure 6 caption: 'metallicty' should be 'metallicity'
L582-584: phrasing is awkward
L594: insert 'it' after NGC 6388?
Figure 7: the lower figure -- it is very difficult to see the red dashed line -- perhaps increase the dash line's width?
L614: 'exceptions' >> 'examples'?
L709: drop the 'a' at the end of the line
L711: insert an 'a' between 'accrete' and 'smaller'
L714: remove 'have been' as the sentence then reads just fine
L718: replace 'from the' with 'on'
L721: remove 'with': 'along the RGB'
L754: have two 'on' at the start of the sentence.
L783: add 's' to star: N-poor stars
L844: remove 'consists' as it exists later in the sentence and makes sense there
L846: 're' should be 'are'?
L937: add 'and' after the comma: 1846, and is
L1052: need 'parameter' or something equivalent after 'responsible'
L1055: to what extent are ...?
L1060: simulations of GCs?
L1075: forth should be fourth
L1076: 'dance' should be 'dances'
L1078: drop 'significantly'
L1084: add 'spectra' after 'low resolution'?
L1087: 'of' should be 'to'?
L1092: add 's' to 'reason'
L1095, 1096: 'turn on' should be 'turn-on'
Author Response
REFEREE 2
This paper is a review of recent results on globular clusters (GCs), specifically reviewing the evidence for multiple populations of stars within a GC. While GCs are not a primary research area for me, I certainly learned a good bit about GCs that I did not know.
Scientifically, I do not have any real quibbles with the paper. However, there are several places in the text where one or more sentences are not clear and these need to be fixed, else the paper will not convey the information that the authors intend.
I do not need to see an updated draft, provided the fixes are made to the text. Throughout my comments, 'L' stands for the paper's Line number.
Authors
We thank the referee for the comments and the corrections that have improved our manuscript. We reply to her/his major comments below. We mark all changes corresponding to her/his significant changes in the paper with boldface text.
We accounted for all suggested minor grammar changes. For clarity, we preferred to not mark them in boldface.
Significant grammar/structure changes:
REFEREE
-- L145: "Since 2P stars ...": this is not a sentence and it is not clear to me what would make it so as I am not certain I understand what the authors intend. The fix *could* be as simple as inserting 'they' after 'oxygen-depleted'.
REPLY
We modified the text as suggested
REFEREE
L200: exactly where is the 'knee'? i did not find a definition within the paper.
REPLY
We indicated in the text the magnitude of the MS knee and define the knee (footnote 2, page 6).
REFEREE
L313-314: the phrase 'To avoid ... supernovae' is not completely clear as currently written. I think the authors are trying to state that you don't want too much heavy element contamination from supernovae, so nature forms black holes. Correct?
REPLY
We tried to clarify the text.
REFEREE
L346: this sentence is not clear: something causes something to be higher than its maximum mass? how is that possible?
REPLY
We clarify that the total amount of the material that is ejected by the super-massive star can be higher than the maximum mass of the star. This is because, according to the scenario by Gieles et al. (2018), the supermassive star, accretes mass in stars that is comparable with its mass loss.
REFEREE
L409-L413: something in this text block is not clear. Certainly, the Cameron-Fowler mechanism should be defined (sure, known to GC researchers, but ...). Perhaps knowing the definition would then make those sentences clear.
REPLY
We added in footonote 4 a short description of the Cameron-Fowler mechanism
REFEREE
Figure 6: uncertainties on the relative flux plot? those two spectra look sufficiently similar to me that i'd think some uncertainties would confirm that.
REPLY
The noise on the continuum is already a strong indicator of the uncertainties of each spectrum. We verified that, based on the errors on the spectra, the two lines have a negligible probability to be the same (p<0.0001).
Clearly, the metallicity difference corresponding to such differences in the equivalent width of the lines mostly depends on stellar parameters.
We prefer not to discuss technical details about the two segments of the spectra plotted in Figure 6 and refer the interested reader to the cited paper by Marino et al. (2009) and Marino (2011) who analyzed the spectra and demonstrated that the [Fe/H] difference between the two stellar populations of this GC has high significance.
REFEREE
L854: 'Helmi stream'?
REPLY
We added a reference to the paper by Helmi et al. (1999) on the discovery of the stellar stream called Helmi stream.
REFEREE
L1036-1039: is this sensible from a physics perspective? gaining mass usually leads to a gain in angular momentum
REPLY
We summarized here the scenario by Wang et al. (2022, Nature Astronomy, in press).
This paper is based on state-of-the-art stellar evolution models that account for stellar rotation and come from the work of the group by N. Langer, who includes the main experts in the field.
We do not have the competence to challenge their models and the way they account for gaining stellar mass and angular momentum.

Reviewer 3 Report
Dear authors,
In my opinion, your review article represents a valuable addition to the field of multiple populations in globular clusters.
I find the text solid and easy to follow. I do have some minor comments, mostly language choice, typos, and omitted references
You are encouraged to consider them.
comments
========
L12: It is not obvious if this stellar mass refers to the host galaxy's or GC's stellar mass
L19: "these stellar fossil records..."
This need to be re-written since the reference in the sentence is to objects rather than data... For example, "They hold stellar fossil records which are used in Galactic ...."
L23: this should be normal case and not title case
L24: It needs to be stated clearly here that SSPs are not the only tools for estimating distances and ages...For example, "...and coeval stars,
from which estimates of distances and ages can be obtained."
L26: For a long time, ...
L34: Galaxy instead of galaxy
L37: remove since and replace with "been"
L43: This statement is unclear to me the way it is written.
I guess a better way of constructing this important sentence could be "... C and O are unique to GCs." or "...C and O are only found in GCs."
L47: "...there has been..."
L48: This is a convenient junction to specify/introduce the acronyms for RGB and SGB
L49: "... dates back to..."
L50: "... sequences which can..."
L51: remove with
L52: write as asymptotic-giant branch (AGB)
L54: Here, and all through the text, M 4 should be written as M4 (without space). Same applies to M 13, M 22, etc all through the text, e.g. L98, L233, L234, L400, L436, L506, L507, L552, L572, L578-605, L666
L699, L754, L870, L913-915
L56: either you decide and stick with multiple population (correct in my opinion) or you amend the entire text (including the title and abstract) with multiple-population
L59: replace "is" with "as"
L61: re-write "has resulted even more complex than" for better clarity
L65: What scenarios (and references)?
L70: Paragraphs on line 65-76 need references! For example, it is not obvious what observational constraints are referenced here? Examples with references would make the text easier to understand!
L73: replace "even" with "... even possibly..."
L74: references for what has been proposed need to be explicitly stated!!!
L77: "... the origin of multiple stellar populations in GCs is not
understood ..." I think this comes off too strong...I strongly suggest that the language be revised to reflect that we do not properly understand the origin of
multiple stellar populations and not that we have no understanding of the phenomenon at all
L94: For a long time, ...
L98: ...revealed that stars...
L103-104: the language used here can be improved...first, elemental abundances are never observed but inferred, and second, the diagrams should be CMDs,
or as you said in the next sentence, photometric diagrams. Please, fix.
L108: point-spread function not point-spread-function
L125: use Str\"{o}mgren
L126: replace "...resulted in..." with "..., is also ..."
Figure~1: Please add the source of these diagrams to the Figure caption. Also, the y-axis labels in the bottom panels need to be properly formatted...
\Delta C_{...}
L145: replace "...have..." with "...they have..."
L150: use "vs. " instead of "against"
L163,169: replace "build" with "built"
L174,175: remove "it has been introduced", and replace band on L175 with "... band has been introduced, ...
L180,181: please re-write these lines for better clarity
L185-187: Since this sentence refers to Integrated photometry as a tool/diagnostic rather than the constructed diagram, I think "diagram" should be replaced with "tool"
L188-192: where are the diagrams in Fig. 1 & 2 sourced from?
L201: use "breadth"
L205: replace "and" with "an"
Figure~2: Please add the source of these diagrams to the Figure caption.
L212: I think a better word choice for "endorse" is "identify" or "define"
L233, 234: missing reference for the works on NGC 6753 and NGC 1851, please fix!
L236: remove almost...or replace with "...are very few or missing."
Figure~3: what are the red dots in the Type-II plot? Also, include the source of these diagrams in the Figure caption
Figure~4: - Please, include the reference for these plots in the caption. Also, properly reformat the y-axis labels \Delta C_{...}. Explicitly specify what MSTO means in the caption
L239-262: I think this section will be significantly improved with a few references.
For example, explicit references of the multi-generation scenarios, the mass-budget phenomenon, and some of the alternative scenarios would be useful.
L240: Astronomers have developed
L242: "Most scenarios comprise two main groups" - I'm not really sure what this sentence means
L244: "...eventually..." not convinced that this is the correct word to use here, kindly fix!
L269: use \textit{p}-capture here and all through the text. Also L320, 333, 445, 784
L270: replace "leds" with "leads"
L280: use "loss" instead of "lost"
L282: should be "...2P stars, which are initially ...". Also replace "...is largely" with "...are largely"
L288: lose instead of loose
L289: remove "...a..."
L290: occur instead of occurs
L291: fix typo in "sthe"
L294: This sentence is incomplete. I think you should add " ... from which the 2P stars are eventually formed" to the end of this sentence. This is because this section is centered on the various formation scenarios of the multiple stellar populations and not just the source of the polluted materials.
L298: "same patterns" should be same abundance patterns???
L304: FRMBs is undefined...I assume this is actually FRMSs
L313: Use "To avoid a situation where the material from which 2P stars form ..."
L319: What follows a runaway star formation? You need to explicitly state this!
L323: "...that, in absence of feedback, would..."
L326: [81] should be Denissenkov and Hartwick [81]. Also, replace "...as the responsible..." with "...as being responsible..."
L327: replace 2P with 2P stars
L329: Use "super-massive" for consistency
L335: remove "the"
L344: use accreted and the protostars
L345: accrete
L348: remove period
L350: possibly instead of possible. Also, use "multiple stellar populations in GCs" instead of "... multiple populations"
L351: use "...half of the massive ..."
L355: replace "...the 2P forms..." with "the 2P stars form from..."
L361: "2P stars are distinctive features of GCs." I guess you meant to say that "2P stars are distinct/unique features of GCs", i.e., 2P stars are only observed in GCs.
L364: For clarity and a more pleasurable reading experience, you need to explicitly state the intended phenomenon, i.e., multiple stellar population phenomenon in GCs! Also, replace "their" with "the"
L368: replace "are feasible" with "...are only feasible..."
L371: replace "...the helium" with "...the total helium abundance."
L375: The results showed that...
L383: remove "...other"
L386: replace "in" with "in the"
L400: NGC 362 is not in references [100,101], only M4, M5, and M12
L401: this is M5 already referenced (assuming [100,101] are the correct references), so you need to fix the phrase "...other clusters..." to exclude NGC 5904. Also, "lover" should be "lower"
L407: add temperature unit to the value
L416: "than the..." should be "compared to 1P stars"
L420: replace "...with respective of the" with "compared to 1P stars"
L424: replace "alone" with "only"
L432: A an should be "As an..."
L458: use "...the only sources responsible..."
L467-478: Lines 467-478 referred to a lot of studies from the literature without any citation. Please fix.
L470-472: Indeed, more massive systems, such as galaxies, have deeper potential wells, required to retain the fast ejecta from Supernovae.
L479: All other title texts are italicized, could you please fix all title text to have the same format and font type?
L486: replace "...from 1P..." with "...from which 1P"
L489-490: at odds reads better than in contrast
L490: come should be came
L500: replace "an" with "the"
L501: replace "...the only responsible..." with "...the only factor responsible..."
L501-503: Why not re-write this sentence with more emphasis on the result rather than the assumption, e.g., "It is possible to take advantage of the 1P F275W − F814W color extension of the RGB
[24] to constrain the internal iron spread by assuming that metallicity spread is the only factor responsible for the color width of 1P stars.
L515: collapse [123].
L519: replace "are much narrow" with "...are narrower"
L530: remove "when"
L531: now has
L532: replace "...position..." with "...separation..."
L551: \textit{s}-elements here and elsewhere. Also, L562, 573, 672, 673, 784
L552: replace "experienced" with "shows", "variations" with "variation", and "than" with "compared to"
L554: comma after GCs
L575-577: Do we have observational evidence that this feature is prevalent in other Type II GCs (apart from M22)?
If so, kindly list them (including references) here. If not, then this sentence needs a re-write to reflect this fact.
Figure~6: s- in "s-rich/Fe-rich and s-poor/" should be \textit{s}-
L579: replace "to" with "with"
L607: replace "hundreds" with "several"
Figure~7: ChMs in the figure caption should not be abbreviated
L619-638: e.g. should be e.g.,
L623: remove (
L632: replace "... in between..." with "...in between the clumps, ..."
L647: remove hyphen in Milky-way
L660: fix formatting of [right panel of Figure8 93]
L669-670: replace "corresponds the" with "corresponds to the"
L681: Please explicitly specify the relations (with references) mentioned here
L682: What are the recent works mentioned here?
L708: replace "... responsible for" with "... mechanism responsible for ..."
L714: remove "have been"
L716: use ... such that 2P stars originated in the dense and compact regions of the cluster center, ...
L718: replace from with on
L719: remove period
L721: remove "with" in "... along with the RGB"
L723: remove "...the" from "...of at least the..."
L739: [26] should be "where Lagioia et al. [26]"
L742: remove "that is"
L748: replace "detected only one out" with "detected that only one out of ...."
L750: move "respectively" to the end of the sentence
L752: replace "...are..." with "...have been..."
L760: replace "... in high..." with "... in the high ..."
L773: use "... stars are born..."
L779: use "Similar to ..."
L786: Use "As with the ..."
L791: Use "The result is that ... "
L793-794, 800-801: re-write these lines for better clarity
L802: I guess the notion refers to the evidence that ancient GCs host multiple populations which must have formed at high redshifts...This is not clear and obvious from the text
L803: replace nitrogen with "nitrogen abundances"???
L809: replace "alone" with "only"
L818: replace "...2P of ..." with "2P stars in ..."
L831: the most remarkable ...
L838: out of 14 ...
L840: accreted not accreated
L844: remove consists
L845: replace "in" with "of"
L846: replace "GCs re" with "GCs that are"
L847: remove which
L858: replace "...on" and "allow shedding light" with "...of" and "... allow us to shed some light ...", respectively
L859: replace "abundance metal-rich ..." with "abundance, metal-rich ..."
L864: replace "have the red HB alone" with "...only have/host red HBs"
L865: replace "indicates" with "is based on"
L866: remove "yet"
L867: write as candidate
L873: write Lirae as Lyrae
L876: replace "resulted" with "is"
L882: replace "has been first..." with "was initially..."
L890: use "helium abundance determinations..."
L892: that the enhanced ...
L895: use "insights" instead of "inside"
L899: use "works" instead of "work"
L903: use "helium variations"
L905: replace "that the 2P" with "they"
L918: replace "allowed to" with "allowed us to"
L923: replace "is a possibly" with "is possibly a"
L928: replace "comprise" with "are"
L937: Split line into two sentences. Introduce the full stop after NGC~1846, and start a new sentence at "It is a universal ..."
L955: H\alpha (no hyphen)
L963: for consistency, this should be "...split MSs ...
L971: for consistency and a better reading experience, change 0.15 to 15%, also change "arises" to "rises"
L975: replace "from firm" with "from making firm"
L978: replace "revealed" with "shown"
L980: replace "with" with "as"
L1023: replace "age rotation" with "age and rotation"
L1026: replace rule with role
L1029: replace [247] with D’Antona et al. [247]. Also, "stars born" should be "stars are born"
L1040: remove period
L1052: replace "only responsible" with "only factor responsible"
L1055: replace "extent star-to-star" with "extent do star-to-star"
L1065: feel is too informal, I'll rather use speculate or opine
L1060-1069: this important needs some references from the literature
L1078: remove significantly
L1081: replace "comprise a minority" with "... is a small fraction ..."
L1087: replace disentangle with disentangling
L1092: replace "reason that are responsible ..." with "reasons responsible ..."
L1096: replace "turn on an ..." with "turn on is an...". Also, replace "clock to" with "clock that can be used to precisely...."
L1101: replace "allow to..." with "allow us to..."
L1116: reference should end with ...4598-2_19 (remove the back slash)
Author Response
We are grateful to the referee for the accurate work and for the many corrections and suggestions that have greatly improved our paper.
We think/hope that we properly accounted for all comments but one. For clarity reasons, due to the large number of changes we preferred to not mark them in the paper.
We did not address the comment on Figure 1:
Also, the y-axis labels in the bottom panels need to be properly formatted...
\Delta C_{...}
and we keep the lower case \Delta_{C...} for consistency with the previous works on the chromosome maps (see e.g. Milone et al. 2017).
